# Targeted mechanical stimulation via magnetic nanoparticles guides in vitro tissue development

Abdel Rahman Abdel Fattah [1,2] ✉, Niko Kolaitis[1], Katrien Van Daele[1], Brian Daza [1], Andika Gregorius Rustandi[1] & Adrian Ranga [1] ✉

Tissues take shape through a series of morphogenetic movements guided by local cell-scale mechanical forces. Current in vitro approaches to recapitulate tissue mechanics rely on uncontrolled self-organization or on the imposition of extrinsic and homogenous forces using matrix or instrument-driven stimulation, thereby failing to recapitulate highly localized and spatially varying forces. Here we develop a method for targeted mechanical stimulation of organoids using embedded magnetic nanoparticles. We show that magnetic clusters within organoids can be produced by sequential aggregation of magnetically labeled and non-labeled human pluripotent stem cells. These clusters impose local mechanical forces on the surrounding cells in response to applied magnetic fields. We show that precise, spatially defined actuation provides short-term mechanical tissue perturbations as well as long-term cytoskeleton remodeling in these organoids, which we term "magnetoids". We demonstrate that targeted magnetic nanoparticle-driven actuation guides asymmetric tissue growth and proliferation, leading to enhanced patterning in human neural magnetoids. This approach, enabled by nanoparticle technology, allows for precise and locally controllable mechanical actuation in human neural tube organoids, and could be widely applicable to interrogate the role of local mechanotransduction in developmental and disease model systems.

During embryogenesis, tissues are generated by morphogenetic events which are tightly controlled by temporally defined mechanical forces. Forces can be generated by intrinsic cellular contractility or as a consequence of adjacent tissues interacting with each other[1,2]. Cells respond to such forces through the activation of mechanosensitive pathways[3,4], and this mechano-responsive feedback serves to change their shape, proliferation, and fate specification[5]. Intrinsic forces generated by tissues undergoing morphogenesis play an important tissue-shaping role. For example, during the development of the early central nervous system, the formation of the neural tube has been attributed to neural plate folding independently of surrounding tissues[6,7]. Indeed, the initiation of neural plate folding has been attributed to local apical constriction at the median hinge point[8] which serves to control floor plate cell shape and cell cycle[9,10]. While it is well established that the response of cellular subpopulations within a tissue to local forces is necessary for the engagement of morphogenetic programs leading to complex tissue architecture, how such forces generate local changes in cell identity and how such responses coordinate tissue-wide growth and patterning effects remains unclear.

Because they can partially recapitulate in vivo cellular diversity and architectural complexity, organoids offer exciting model systems

[1]Laboratory of Bioengineering and Morphogenesis, Biomechanics Section, Department of Mechanical Engineering, KU Leuven, Leuven, Belgium. [2]Present address: CeMM Research Center for Molecular Medicine of the Austrian Academy of Sciences, Vienna, Austria. ✉e-mail: aabdelfattah@cemm.oeaw.ac.at; adrian.ranga@kuleuven.be

to help address mechanobiological questions while maintaining control over the mechanical and biochemical microenvironments. The use of synthetic extracellular matrices such as polyethylene glycol (PEG) hydrogels has shown that matrix stiffness modulation can not only sustain organoid cell growth and morphogenesis[11], but also guide coordinated multicellular differentiation[12] and patterning[13]. Such studies rely primarily on predefined gel stiffnesses or matrix degradation, where the mechanical characteristics of the microenvironment cannot be actively modified. Dynamic matrices allow for changes in matrix properties, for example by light-induced crosslinking to control axon projections in spinal cord organoids[14] or by matrix rearrangement via reversible hydrogen bonding to promote crypt formation in intestinal organoids[15]. However, such matrices do not impose active forces on the tissues, thereby failing to recapitulate dynamic mechanical events seen in vivo. To impose active mechanical perturbations to organoid cultures, we previously developed a mechanical device which could control force modulation on organoids, and demonstrated an enhancement of floor plate patterning in organoids upon actuation[16]. This approach can only impose external and largely homogeneous forces on tissues, allowing an understanding of mechanoregulatory effects of global forces but not local ones.

Various approaches have been implemented to mechanically stimulate tissues locally. Optical tweezing can deliver subcellular point forces but is limited to piconewton ranges and thus to applications requiring only weak forces, such as the mechanical actuation of ion channels[17] and protein complexes related to synaptic activity[18] in neurons. Larger forces can be delivered using techniques such as microindentation[19] and micro-aspiration, which have been used in a variety of applications such as in elucidating electrical-mechanical coupling in retinal cells[20] and in uncovering a biophysical mechanism involved in the regeneration of mucociliated epithelium[21]. While able to deliver local forces, such techniques require tissue-surface contact and are thus limited to stimulating external forces. In three-dimensional developing tissues, force generation is not limited to the surface, and the ability to provide internal mechanical stimulation is therefore needed.

To mechanically stimulate tissues internally, magnetic nanoparticles (MNPs) have been used: once embedded in a tissue, they can respond to a magnetic field and cause localized mechanical stresses. For example, ferrofluids embedded in zebrafish embryos have been used to demonstrate region-specific tissue stiffness[22]. This technique is well suited to in vivo investigations, however, due to the relatively large size of the magnetic actuators (-100 μm), is less applicable for smaller in-vitro model systems. MNP internalization offers an alternative that allows for cell-level in-vitro actuation. Internalized MNPs have been used to demonstrate polarized increases in cytoskeleton rearrangement[23] in response to mechanical stresses from an external magnetic field. This type of remote actuation is also scalable to multicellular constructs. For example, microfabricated magnetic patterns have been used to impose cyclic mechanical forces on embryonic stem cell aggregates labeled with magnetic nanoparticles, providing better cardiac differentiation and leading to higher efficiency of beating cellular constructs[24].

While demonstrating the power of magnetic, remote, and local actuation in-vitro, these studies have been limited to inducing local forces in 2D or to imposing global forces in 3D. There has thus far not been any way to impose more in vivo relevant local forces within organoids, and the corresponding mechanoregulatory effects have therefore remained largely underexplored. Here, we develop an approach to actuate local regions within an organoid using embedded magnetic nanoparticles, which we term magnetoids, and show that global magnetic fields can be used to generate internal and local forces which modulate growth, morphogenesis and patterning within the organoid body.

## Results

### Magnetoid generation

Our approach aimed at providing localized forces by concentrating magnetic nanoparticle into clusters (MagCs) within a region of an organoid. To create a magnetoid, we developed a simple protocol which ensured that under a magnetic field, only the tissue in close proximity to the MagC would be mechanically actuated in response to the magnetic field, thus providing local mechanical forces.

In order to visualize the location of MagCs by fluorescence microscopy, we first labeled MagCs with carboxylated fluorescent particles (FPs) (Fig. 1a). The fluorescent MagCs were then incubated with single hPSCs to magnetize the cells. For maximal cell magnetization, a high concentration of NPs is desirable, however, this may be deleterious to cell viability[25–29]. To optimize the concentration of MNPs, we incubated hPSCs with a range of MNP concentrations, and determined that 1000 μg/ml was the highest MNP concentration which would maintain a high cell viability (Fig. 1b). In contrast with most previous uses of MNPs, here our MNP clusters do not undergo uptake through endocytosis due to their relatively large size (-2 μm) (Supplementary Fig. 1a), and are instead adsorbed on the membrane exterior and found primarily on top (apical side) of the hPSC colonies, as demonstrated by co-localization with the expression of ZO1 (Supplementary Fig. 1b).

We next verified that the MagCs adhered to the cells to produce magnetized hPSC (mhPSCs), and found that the adhesion was strong enough to guide cells by magnetophoresis in a liquid medium where nearly 100% of cells display mobility, i.e. cells were successfully magnetized (Fig. 1c). Of note, the incubation of MagCs with cells was necessary since instant mixing of MagCs with single hPSCs did not provide adequate membrane adsorption of the particles (Fig. 1c). To estimate the volume of adsorbed MagCs per cell, a small static magnet was used to move single mhPSCs in suspension in a growth medium (Fig. 1d). Due to Stoke's drag conditions, the magnetic force was evaluated as equal to the calculated drag force (Fig. 1e), and the MagC volume was estimated to be -7.3 × 10³ μm³ per cell (see methods, Fig. 1f).

Magnetized pluripotent aggregates with localized MagCs (Fig. 1a) were produced by first centrifuging mhPSCs followed by a second centrifugation step to aggregate the non-magnetic hPCSs. This consecutive centrifugation allows mhPSCs and their MagC tags to collect together at the bottom of the aggregation well. We subsequently perform a 1-day incubation under a magnetic field. Because the MagCs are only transiently attached to their cellular chaperones, once they become magnetized under the magnetic field, a force is generated within the MagCs and they leave the cell surfaces in an attempt to travel towards the magnet. However, because MagCs are in close proximity to each other, they undergo directed assembly as they interact and influence each other through their own induced magnetic fields, allowing them to build rod-shaped MagCs inside the organoids, thereby ensuring force localization. Instantly mixing magnetic nanoparticles with non-magnetized hPSCs produced magnetoids with multiple MagC clusters and thus force foci, an undesirable effect that diffuses the force rather than localizes it. These magnetized pluripotent aggregates were then embedded in PEG hydrogels with a stiffness of 2 kPa as previously described[16], and could then be differentiated under constant or periodic magnetic actuation to impose internal local forces through the displacement of the rod-shaped MagCs.

To investigate the forces generated by the MagCs, we generated human neural tube magnetoids (hNTmOs) following a previously described human neural tube differentiation protocol[16]. We first evaluated the magnetic field produced by a static magnet placed adjacent to a standard well plate. We found that magnetoids situated furthest from the magnet surface were subjected to the lowest forces (Supplementary Fig. 2a). While these low forces led to MagC fragmentation, higher forces closer to the magnet led to high MagC aspect ratios and

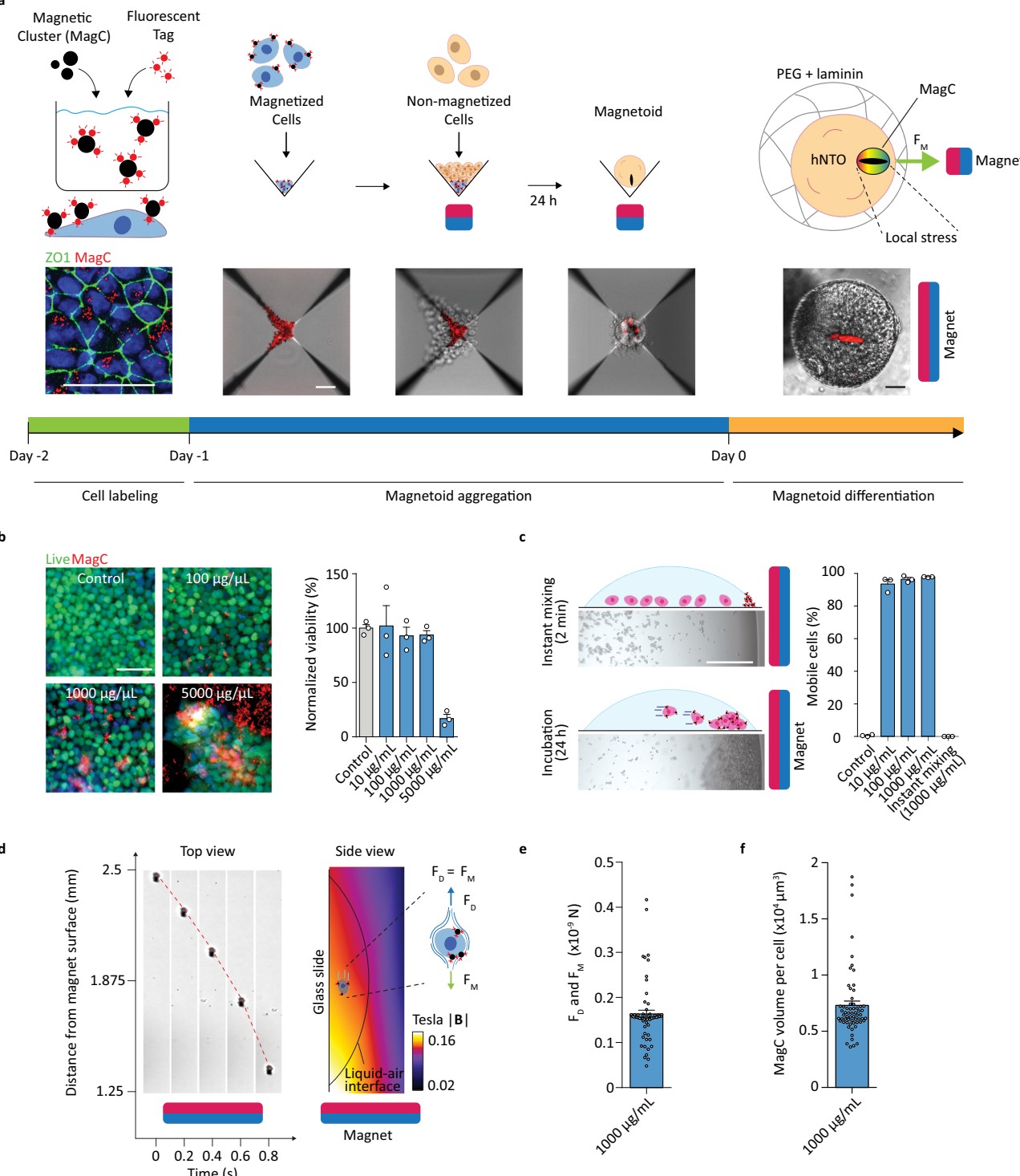

**Fig. 1 | Human PSC magnetization and magnetoid generation. a** Magnetoid generation and hNTmO differentiation protocol. Scalebars 50 μm. **b** Toxicity test with normalized hPSC count after 3-day exposure to various concentrations of MagCs (*n* = 3 independent experiments). Representative images showing live hPSCs in control and MagC conditions (100, 1000, and 5000 μg/mL) (n = 3 independent experiments). Scalebar 50 μm. **c** Representative images showing mhPSC magnetic attraction (*n* = 3 independent experiments). Percentage of mobile cells at various MagC concentrations (*n* = 3 independent experiments). Scalebar 500 μm. **d** Representative images of a mhPSC undergoing magnetophoresis. Corresponding magnetic field as simulated by a 3 × 3 × 3 mm N45 magnet (*n* = 3 independent experiments). Schematic representation of Stoke's drag and magnetic forces. **e** Stoke's drag and magnetic forces on mhPSCs undergoing magnetophoresis, and (**f**) corresponding MagC volumes (*n* = 3 independent experiments for 62 cells). Error bars are SEM. Source data are provided as a Source Data file.

maintained cluster localization (Supplementary Fig. 2b, c). Thus, to maintain consistent force localization and highest force values in our investigation, magnetoids were placed in wells closest to the magnet surface.

We next investigated the effect of initial mhPSC proportion on force generation. We found that increasing the initial mhPSC proportion increased MagC volume (Fig. 2a) and by extension the resultant force (Fig. 2b). To ensure the embedded MagCs do not

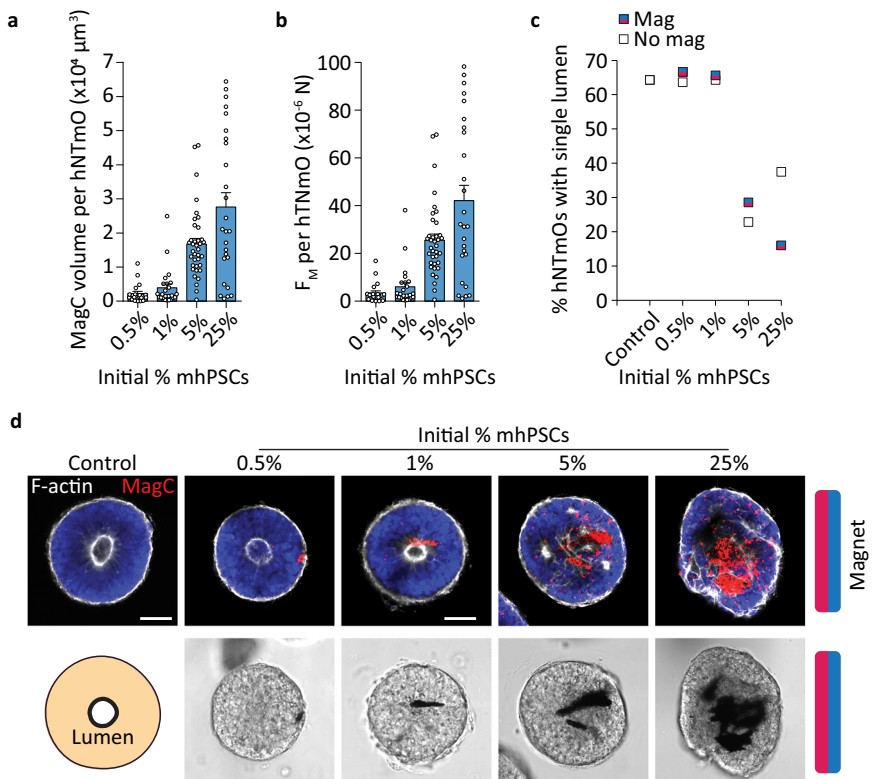

**Fig. 2 | Magnetoid force estimation and cytoskeletal evaluation. a** Estimation of MagC volumes in day 11 hNTmOs at various initial mhPSC proportions ($n = 3$ independent experiments for 21 (0.5%), 24 (1%), 37 (5%), 26 (25%) hNTmOs). **b** Estimation of force generation in day 11 hNTmOs at various initial mhPSC proportions (from **a**). **c** Average lumen formation in day 11 hNTmOs at various conditions and with various initial mhPSC proportions ($n = 2$ independent experiments for >30 magnetoid per condition). **d** Representative images showing F-actin expressions in day 11 hNTmOs at various initial mhPSC proportions ($n = 3$ independent experiments). Scalebar 50 μm. Error bars are SEM. Source data are provided as a Source Data file.

disrupt the magnetoid cytoarchitecture, an important consideration for correct differentiation and growth[13,16], we investigated the cytoskeleton arrangement in day 11 hNTmOs with varying initial mhPSC proportions. Specifically, we reasoned that the formation of a single lumen within the organoid body would be a good indicator of cytoskeletal health, and based our assessment of correct cytoarchitecture on the fraction of hNTmOs with single lumens (Fig. 2c, d). On average, low MagC volumes of ~$5.7 \times 10^3$ μm³ correlated with single lumen magnetoids, indicating correct apicobasal polarization (Supplementary Fig. 3), which can only result if cell–cell interactions are not significantly impeded. Indeed, only the low initial mhPSC proportions (≤1%) that produce small MagC volumes resulted in single lumen magnetoids with the same frequency as the MagC-free control organoids. In contrast, proportions >1% reduced single lumen formation frequency regardless of magnetic field application (Fig. 2c–d). Thus while 5% and 25% mhPSCs-magnetoids generated more force, the 1% mhPSCs magnetoid condition was chosen in our study as it provided the maximum force (~$6.0 \times 10^{-6}$ N) with minimal impact on organoid cytoskeletal organization and cell–cell interaction. This force was also found to be in the range of those previously reported to stall neural tube folding in amphibia[30], and suggests that the magnetoid model system is able to attain relevant in vivo forces. While precise a priori control over MagC position within the magnetoids is not possible, we found that MagC positions under 1% mhPSC-magnetoids were centered and biased toward the magnet position (Supplementary Fig. 4), thereby providing a predictable and reproducible MagC localization.

Overall, these results demonstrate that magnetoids can be created with localized MagCs that are responsive to an external magnetic field while maintaining organoid cytoarchitecture. While forces can be simply generated by a static magnet, their strength and timing affect the MagC-magnetoid interactions. We therefore thought to explore various force profiles to evaluate such interactions.

## Short and long-term magnetic actuation in magnetoids

The temporal dynamics of force applications are important in studying the mechanobiology of cellular processes, which range from seconds and minutes, such as in heart contractions, or days for developmental events such as neural tube morphogenesis. We therefore thought to investigate the applicability of the magnetoid model system in emulating various force dynamics regimes. We evaluated the magnetoids' response to magnetic actuation by assessing tissue displacement and cytoskeleton rearrangement.

To evaluate the short-term response of organoids to fast-changing forces, an actuation burst was provided by introducing a magnetic field to a magnetoid, which was observed over 30 s, followed by magnet removal. This short perturbation event is far below reported force application durations (~1 h) needed for cytoskeletal reorganization[31]. Upon magnetic field exposure, the embedded MagCs attempt to align their long axis with the magnetic field lines, pushing against the surrounding tissue and providing actuation (Fig. 3a, b). Since small variations occur in the displacement of different MagCs when subjected to a magnetic field, for comparison purposes we normalized the tissue displacement map to the distance traveled by the corresponding MagC, $d_M$, upon actuation. We found that tissues positioned >25 $d_M$ from the MagC undergo minimal displacement (Fig. 3c). Since the size of magnetoids is on the order of ~100 $d_M$ these results suggest that magnetoids can be used to deliver fast-changing localized forces actuating only the tissue in close proximity, leaving the remainder of the magnetoid unaffected.

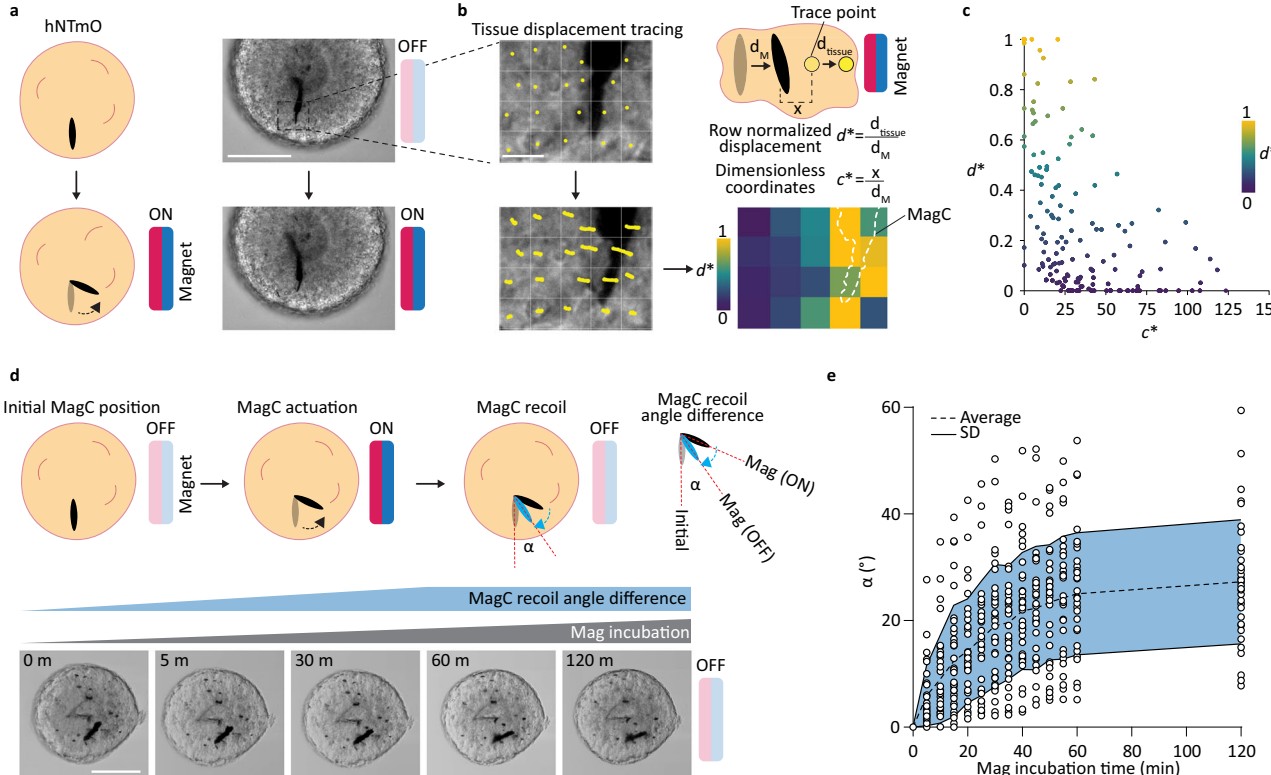

**Fig. 3 | Short- and long-term local magnetic actuation. a** Schematic representation of local short-term actuation of MagCs. Representative images of day 11-hNTmOs with and without the influence of magnetic field ($n = 3$ independent experiments for a total of 40 hNTmOs). Scalebar 50 μm. **b** Schematic representation of tissue displacement. Representative images of tissue tracing measurements and accompanied heatmap ($n = 3$ independent experiments for a total of 40 hNTmOs) Scalebar 10 μm. **c** Correlation of dimensionless tissue and MagC displacements ($n = 3$ independent experiments for a total of 20 hNTmOs). **d** Schematic representation of MagC actuation and recoil (gray oval: MagC initial position, black oval: MagC position under a magnetic field, blue oval: MagC final position after removal of magnetic field). Representative images of MagC actuation and recoil during long-term magnetic incubation. ($n = 4$ independent experiments for 35 hNTmOs). Scalebar 50 μm. **e** Recoil angle difference α as a function of magnetic incubation time ($n = 4$ independent experiments for 35 hNTmOs). Dashed line represents the average α values and solid lines represent SD. Source data are provided as a Source Data file.

Slow-changing forces can also be generated by subjecting magnetoids to constant magnetic fields for longer time periods. In order to investigate the transition from fast (bursts) to slow (constant) forces, we evaluated the MagCs recoil angle difference, α, which is the angular shift between the original (Fig. 3d gray oval) and final (Fig. 3d blue oval) MagC position upon recoil from the magnetized position (Fig. 3d black oval) after removal of the magnetic field (Fig. 3d). Short actuation times led MagCs to recoil to their original orientation once the magnetic field was lifted (Fig. 3e). In contrast, longer actuation under a magnetic field led to higher values of α that stabilized after 1 h. These results suggest that constant forces applied over longer durations likely trigger cellular reorganization within the vicinity of the MagC as it aligns its body axis according to the new magnetic field lines. This reorganization prevents the MagC from returning to its original position once the magnetic field is lifted. More importantly, once realigned, the MagC continues to apply a localized force to its neighboring cells. This mode of actuation is most suitable for relatively slow tissue deformations. Although they do not recapitulate the complexity of the neural tube, we use magnetoids under constant magnetic actuation as a minimal mechano-biological model to study the effect of local forces on neural tube morphogenesis and patterning.

### Local magnetic actuation biases growth and proliferation in human neural tube magnetoids

Neural tube development relies on differential cell cycle regulation and growth[32] where floor plate cells[9,10] proliferate less than intermediate domains[33]. To investigate whether differential forces can guide these processes, we first investigated morphological changes in hNTmOs. Organoids without applied magnetic field, with or without MagCs, were seen to grow symmetrically with an average $P_m/P_a$ ~ 1.05 and ~1.01, respectively (Fig. 4a), where $P_m$ and $P_a$ are the measured linear growth toward and away from the magnet respectively. In contrast, constant magnetic actuation over 11 days biased growth away from the magnet ($P_m/P_a$ ~ 0.61). Despite this growth bias, we observed no significant differences in hNTmO sizes between all three conditions (Supplementary Fig. 5), suggesting that magnetic forces do not change the overall proliferation capacity of organoids but only bias them directionally. To determine whether the observed growth bias was due to differential proliferation or internal deformation of the hNTmOs, we performed EdU staining. Proliferation appeared largely similar in the absence of a magnetic field, where the ratio of EdU intensity between the regions facing the magnet ($EdU_m$), and those facing away from the magnet ($EdU_a$), resulted in a $EdU_m/EdU_a$ ~ 1.28 and ~1.55 in magnetoids with and without MagCs respectively (Fig. 4b). However, application of the magnetic field produced a significant proliferation bias away from the magnet with $EdU_m/EdU_a$ ~ 0.80. These results suggest that after the initial realignment of the MagCs under magnetic incubation conditions, causing cellular reorganization (Fig. 3b, c), the subsequent effect of the sustained magnetic force is to modulate growth and proliferation. Indeed, MagCs likely compress the tissue facing the magnet, thereby counteracting growth in this region, while further away from the MagC position, unrestricted tissue can freely proliferate. Zones of reduced proliferation also reduced the cell density in the hNTmO

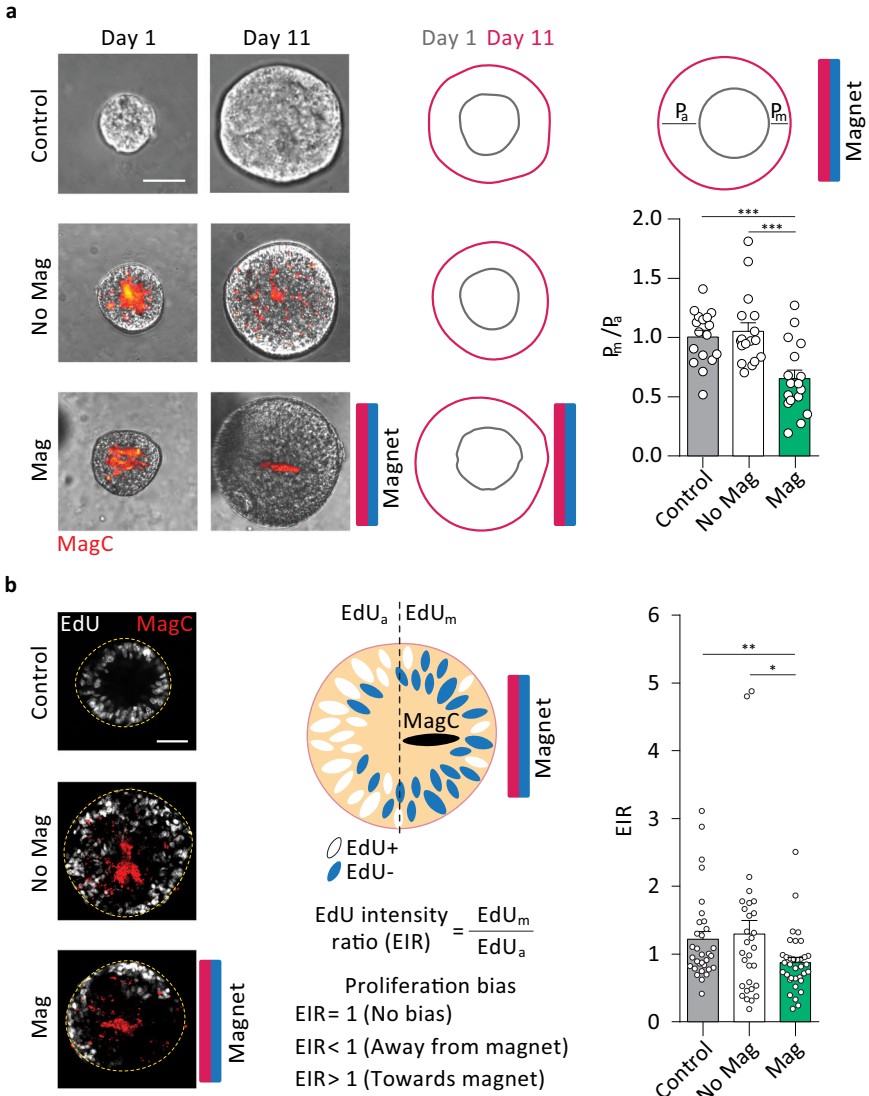

**Fig. 4 | Local magnetic actuation biases growth and proliferation.**
**a** Representative images of day 1 and day 11 hNTmOs under various actuation conditions (n = 3 independent experiments). Schematic representation of day 1 and day 11 hNTmO contours. Growth ratio of hNTmOs for various actuation conditions (n = 3 independent experiments for a total of 17 hNTmOs per condition). Statistical analysis was performed by unpaired two-sided t test. Control-Mag p = 0.0005, No Mag-Mag p = 0.0005. **b** Representative images for EdU staining of day 11 hNTmOs

under various actuation conditions (n = 2 independent experiments). Schematic representation of biased proliferation in hNTmO under the influence of magnetic field. Proliferation ratio of day 11 hNTmOs for various actuation conditions (n = 3 independent experiments for 32 (Control), 30 (No Mag) and 35 (Mag) hNTmOs). Statistical analysis was performed by unpaired two-sided t-test. Control-Mag p = 0.0077, No Mag-Mag p = 0.0176. Scalebars 50 μm. Error bars are SEM. Source data are provided as a Source Data file.

halves that are facing the magnet, as measured by the lower averaged Hoechst signal in these halves, compared to those facing away from the magnet resulting in an average Hoechst intensity ratio (HIR) < 1 (Supplementary Fig. 6). In contrast, in control organoids and hNTmOs not subjected to a magnetic field, cell densities on either half were comparable with HIR ~ 1. This result suggests that local compressive forces serve to decrease cell proliferation and cell density in these regions.

Local actuation can therefore modulate local cell cycle to bias global growth direction. Increased cell cycle and proliferation have been reported in the intermediate domains of the developing mouse neural tube in comparison to ventral domains[33], where the floor plate cells remains non-proliferative[13]. Moreover, mechanical stresses have been shown to guide cell fate specification and patterning in mouse NTOs[13,34]. We have previously shown that global mechanical stretching increased cell proliferation in human NTOs, concomitant with a reduction in FOXA2+ cells, thereby leading to enhanced patterning of

this subpopulation[16]. Taken together, these results suggest that mechanics-induced modulation of cell cycle in hNTmOs could be linked to changes in fate specification and patterning. We therefore thought to investigate whether the observed changes in local proliferation in our hNTmOs were linked to mechanically-driven biases in patterning.

## Local magnetic actuation guides tissue-wide patterning in human neural tube magnetoids

Magnetoids were differentiated towards the neural fate upon embedding in the PEG hydrogels, using retinoic acid (RA) and smoothened agonist (SAG) treatment from days 3 to 5 to posteriorize and ventralize emergent cell fates. The magnetic field was applied to the magnetoids for the entire 11 day duration of the experiment. We first investigated the effect of the magnetic field alone on the induction of ventral and intermediate neural tube domains by analyzing the expression of FOXA2 and PAX6 respectively. When subjecting

MagC-free organoids to a magnetic field, we found that cytoarchitecture and frequency of FOXA2 and PAX6 were similar to control conditions not subjected to a magnetic field, indicating that magnetic fields alone do not alter the biological responses in this model system (Supplementary Fig. 7). We further expanded our analysis to assess other differentially regulated regions in the neural tube, including ventral domain markers NKX2.2, OLIG2 and NKX6.1, motor neuron marker ISL1 and neuronal marker TUBB3. We observed no significant changes in the induction frequency of any of the fates in either actuated or unactuated conditions with MagCs compared to controls without MagCs, indicating that the presence of MagCs does not selectively abrogate specific fates in our model system (Fig. 5a). Upon magnetic actuation, FOXA2+, NKX6.1+, OLIG2+ and PAX6+ hNTmOs all exhibited an increase in patterning frequency compared to either control organoids and unactuated hNTmOs. We further assessed the occurrence of intermediate and ventral fates within the same organoid and found a higher frequency of PAX6+/NKX6.1+ double positive organoids in actuated conditions (26.2%) compared to control (12.5%) and unactuated hNTmOs (15.8%) (Supplementary Fig. 8), underscoring the enhanced patterning as a result of magnetic actuation.

We next assessed whether a directional patterning bias occurred in actuated hNTmOs with reference to the magnet position. For each marker, we projected the fate expressions from all observed organoids onto the same space, scaling respective organoid size where necessary to ensure proper mapping while keeping track of the magnet position (Fig. 5b). Strikingly, strong polarization of FOXA2, NKX6.1 and PAX6 was observed in actuated organoids, with preferential expression towards the magnet position for FOXA2 and NKX6.1 and away from the magnet position for PAX6. NKX6.1 expression appeared more expansive than FOXA2 floor plate expression, as it known to be more broadly expressed across multiple ventral domains. NKX2.2 and OLIG2 expression was also seen across broader areas (Fig. 5b white arrow heads), appearing laterally to the magnetic force vector. Motor neuron ISL1+ cells, possibly originating from OLIG2+ progenitor motor neuron (pMN) cells, also had high bilateral expressions in actuated hNTmOs (Fig. 5b white arrow heads). In contrast, in control and unactuated hNTmOs, integrated expressions of most fates showed random hotspots with no directional bias.

We further quantified directional patterning bias by evaluating the average fate fluorescence intensities away and towards the magnet (Fig. 5c). We found the most significant differences to be associated with FOXA2, NKX6.1 and PAX6, which are the domains that most polarize towards or away from the magnet. Linking these results to observations of proliferation suggests that floor plate cells pattern in regions with lower proliferation, while those of the intermediate fates are preferentially found in regions with higher proliferation. This domain allocation may then provide the space for other ventral domains to emerge, such as the p3 (NKX2.2) and pMN (OLIG2) domains that occupy lateral positions

To further illustrate that force localization biases not only pattern direction of a single fate, but the organization of multiple patterns and fates, we analyzed integrated FOXA2, NKX6.1, NKX2.2, OLIG2, and PAX6 expression profiles in scaled organoids along various projection angles (Fig. 5d). We then investigated the correlation, at each projection angle, with an ideal pattern that minimally recapitulates the dorsoventral organization of the same domains in the in vivo neural tube. Fates in the control and unactuated hNTmO cases occurred largely at random locations, therefore, as expected, rotation of the organoids did not yield strong correlation with the ideal pattern (Fig. 5d). In the magnetically actuated hNTmOs, a strong correlation with the ideal patten was observed at a rotation angle of 0°, which is the baseline orientation with reference to the magnet position (Fig. 5d). This analysis demonstrated that pattern organization is a function of magnet position and that this organization is not recapitulated in magnet-free and control conditions at any other angle. To better visualize pattern

organization, we merged the top 20% of the integrated fluorescence intensity of FOXA2, NKX6.1, OLIG2, and PAX6 expressions within the same organoid space keeping the original organoid orientation (Fig. 5e). Compared to the control and unactuated hNTmOs, actuated hNTmOs showed clear dorsoventral organization with mutual exclusion of FOXA2, OLIG2 and PAX6 domains in the correct order, with NKX6.1 sharing the domain space with FOXA2 and OLIG2 but not PAX6. Although not all magnetically actuated organoids yield this degree of organization due to variations between organoids, these results demonstrate that fate specification follows a domain allocation program that is determined by the magnetic force acting on the organoid. In addition, the spatially encoded fate specifications we observe as a function of force stimulation suggest that local forces may act as a mechanical frame of reference to orient and guide patterns across the tissue.

To address whether the localized force is necessary to maintain patterning after it is established, we removed the magnetic field at day 9 and assessed FOXA2 expression at day 11, as well as at day 11 and assessed FOXA2 expression at day 13 (Supplementary Fig. 9). In both cases, organoids where mechanical stimulation via magnetic fields was removed at late time points retained a high FOXA2+ patterning efficiency (magnetic field removal at day 9: 57.7%, magnetic field removal at day 11: 62.7%) similar to those actuated continuously (Fig. 5a), compared to the respective unactuated controls. These results indicate that removal of the magnetic field does not significantly affect the pattern once it is established, and suggests that FOXA2+ subpopulations form stable self-sustained patterns that are not hindered by the removal of the mechanical stimulation which initiates the pattern formation.

As the in vivo neural tube increases in size, cell patterning is maintained through antiparallel gradients operating across the growing tissue with cells interpreting ranges of biochemical cues at various distances[35]. With regards to these observations, our magnetoid system allows us to explore a mechanical analogue of this relationship, specifically the length scale of cellular interpretation of a mechanical force gradient. Because not all MagCs have the same volume and not all the organoids are at the same distance away from the magnet surface, the generated local forces are not the same across organoids. The resultant range of forces provided us with an opportunity to examine force magnitude and the distance of cells expressing different fates. By measuring distance between the furthest expressing FOXA2+ or PAX6+ cell to the MagC in actuated magnetoids we can describe the range of the generated local force in influencing cells away from the MagC (Supplementary Fig. 10). Overall, we found that the higher the force, the more distant the expression of the furthest FOXA2+ and PAX6+ cells (Supplementary Fig. 10), indicating that higher forces have larger effective areas in which cells can be influenced to adopt specific fates, and more broadly that the position of cells in relation to the source of the mechanical force is involved in specifying its fate.

## Discussion

Altogether, our results demonstrate that internal local tissue actuation can guide tissue-wide organoid morphology, growth, and patterning. These local mechanical forces are achieved using a simple protocol to embed local magnetic tissue actuators in organoids. We show that this approach can generate both fast and slow-changing forces, and that rapid short-term actuation reversibly deforms local tissue, while longer-term actuation causes permanent reorganization and tissue-scale differential growth. While no preferential direction was previously observed in neural tissue under exogenous actuation[16,36], our results demonstrate that local actuation guides directional patterning bias. Moreover, while local magnetic actuation is demonstrated in human neural tube organoids here, the use of magnetoids in other multicellular contexts could serve to address a wide variety of mechanobiological questions related to the effects of local forces in development and disease.

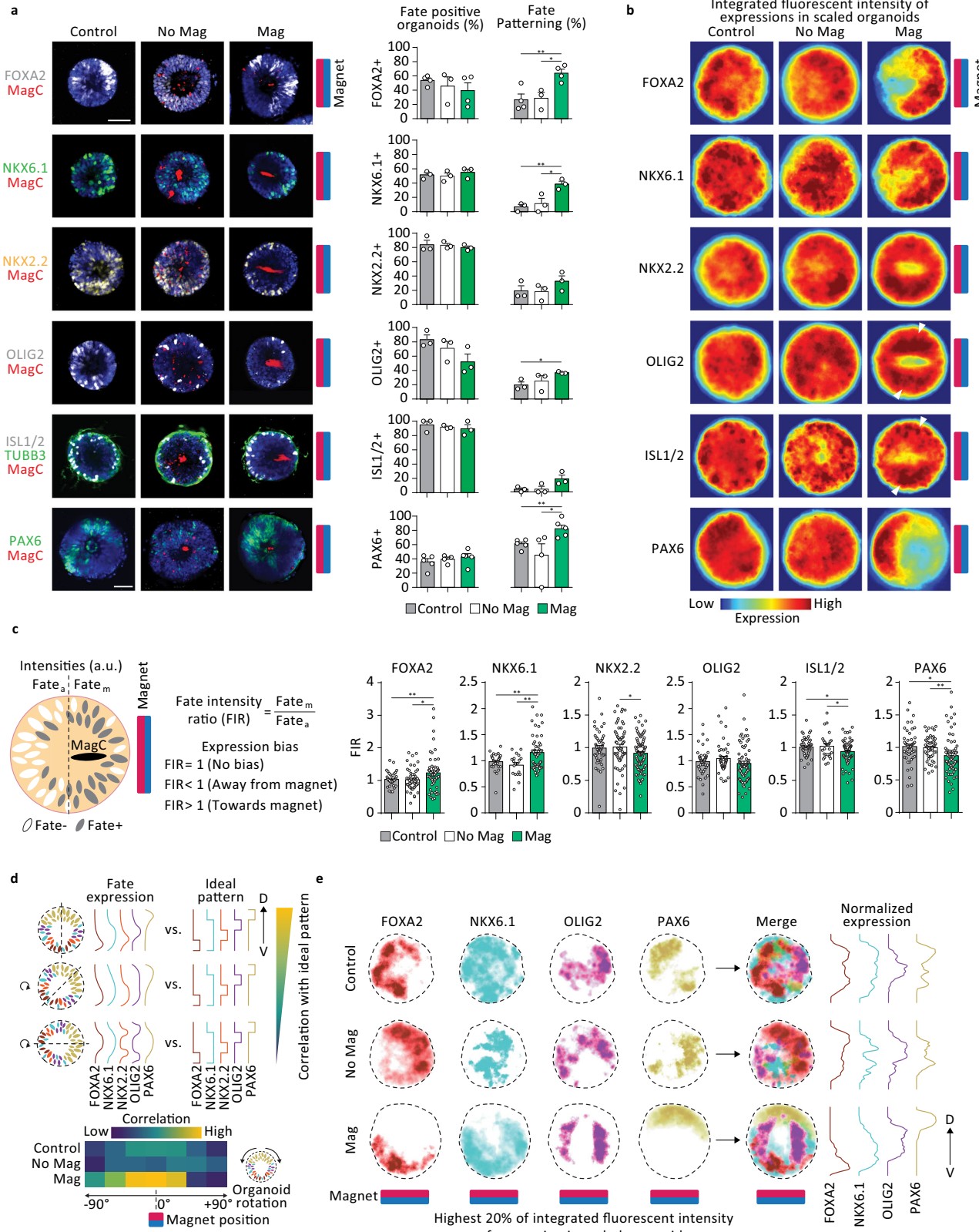

## Methods

### Human PSC lines

The human PSC lines used in this study are 1) NCRM-1 (RRID:CV-CL_1E71) hPSC line from NIH Center for Regenerative Medicine (CRM), Bethesda, USA. 2) ZO1 hPSC line (Mono-allelic mEGFP-Tagged TJP1 WTC, Coriell institute for Medical Research), 3) hPSC

F-actin reporter line provided by Catherine Verfaillie, Stem Cell institute Leuven.

### Human PSC culture

Matrigel-coated 6 well plates were used to maintain hPSCs cultured to 60–70% confluence. Human PSC colonies were passaged using a

**Fig. 5 | Local magnetic actuation biases patterning in hNTmOs. a** Representative images showing FOXA2, NKX6.1, NKX2.2, OLIG2, ISL1, TUBB3, and PAX6 expressions in day 11 hNTmOs under various actuation conditions. Respective quantification of fate induction and patterning in hNTmOs under various actuation condition ($n = 3$ independent experiments for NKX6.1, NKX2.2, OLIG2, ISL1, TUBB3, $n = 4$ independent experiments for FOXA2 except for the No Mag condition $n = 3$ independent experiments, $n = 5$ independent experiments for PAX6 except for the No Mag condition $n = 4$ independent experiments, a total of >60 hNTmOs for all data points per condition). Statistical analysis was performed by unpaired two-sided $t$ test. FOXA2:Mag-Control $p = 0.0092$, FOXA2: Mag-No Mag $p = 0.0143$, NKX6.1:Mag-Control $p = 0.0040$, NKX6.1: Mag-No Mag $p = 0.0317$, OLIG2:Mag-Control $p = 0.0162$, PAX6:Mag-Control $p = 0.085$, PAX6:Mag-No Mag $p = 0.0467$. **b** Integrated fate expressions in scaled organoids, color coded by expression frequency. Hotspots in the resultant heatmaps convey regions with the highest frequency of fate expression across all analyzed organoids compared to cooler regions with less frequent expressions. White arrows denote OLIG2 and ISL1 lateral expression hotspots (data from **a**) **c** Schematic representation of biased fate expressions under magnetic actuation condition. Quantification of FOXA2, NKX6.1, NKX2.2, OLIG2, ISL1, and PAX6 intensity ratio for day 11 control organoids and hNTmOs under various actuation conditions (data from **a**). FOXA2:Mag-Control $p = 0.0074$, FOXA2: Mag-No Mag $p = 0.0104$, NKX6.1:Mag-Control $p = 0.0051$, NKX6.1: Mag-No Mag $p = 0.0039$, NKX2.2: Mag-No Mag $p = 0.0386$, ISL1:Mag-Control $p = 0.0205$, ISL1: Mag-No Mag $p = 0.0488$, PAX6:Mag-Control $p = 0.0423$, PAX6:Mag-No Mag $p = 0.0020$. **d** Schematic representation of organoid rotation and expression profile assessment of domains along the dorsoventral (D-V) axis and their correlation compared to an ideal pattern. Correlation heatmap at various organoid rotation angles color-coded by correlation strength. **e** Highest 20% of integrated fluorescent intensity of fate expression from (**b**) for FOXA2, NKX6.1, OLIG2 and PAX6 and associated expression profiles. Scalebar 50 μm. Error bars are SEM. Source data are provided as a Source Data file.

treatment of Dispase (Sigma) for 4 m at 37 °C followed by PBS washes. 1 mL of Essential 8 (E8)−Flex Medium Kit (ThermoFisher Scientific) supplemented with 1% Penicillin Streptomycin (GIBCO) and Y-27632 Rock inhibitor (ROCKi) (Hellobio) at 10 μM was used to passage colonies after gentle scraping and agitation with a pipette to break down the colonies. Colonies were passaged 1:6 and incubated in 2 mL of E8-Flex medium supplemented with ROCKi at 10 μM for 24 h. The medium was replaced by 4 mL of fresh E8-Flex and the colonies were incubated for an additional 48 h when they reached 60−70% confluence.

## Magnetic nanoparticle (MNP) preparation and magnetic cluster (MagC) labeling with red fluorescent particles

100 mg of Fe3O4 magnetic nanoparticles (MNPs) (Sigma-Aldrich, 20−40 nm) were first weighed under sterile conditions and added to 1 mL of E8-Flex media, in a 1.5 mL Eppendorf tube, resulting in a stock MagC mixture with a concentration of 100 mg/mL. The solution was pipetted thoroughly and subsequently sonicated for 5 m to break apart any aggregation. To label the MagCs, 20 μL (2 mg of MagCs) were taken from the stock mixture, immediately following sonication, and added to 2 mL of E8-Flex medium supplemented with 10 μM of ROCKi. The mixture was then sonicated for 3 m to break up aggregation. Next red fluorescent beads (FluoSpheres Carboxylate-Modified Microspheres, 200 nm, Invitrogen) were diluted 1:10 in E8-Flex supplemented with 10 μM of ROCKi and sonicated for 5 m to break up any aggregation. Subsequently, 5 μL of the diluted fluorescent particles were added to the 2 mL MagC-medium mixture to obtain a final composition of 0.025% v/v (1:4000 dilution of the stock solution). The mixture was immediately sonicated for 3 m to increase MNP-fluorescent particle interaction resulting in efficient labeling of the MagCs. This created a 2× magnetic cluster (MagC) solution.

## Magnetization of human induced pluripotent stem cells

Human hPSCs, at a confluence of 60−70%, were washed with PBS three times, followed by the application of 1 mL of TrypLE Express (GIBCO) at 37 °C for 3 m for dissociation. Once dissociated into single cells, 9 mL of DMEM/F12 medium supplemented with 20% FBS (GIBCO) was added for neutralization and TrypLE Express washing. The cells were then centrifuged at 300 RCF for 3 m After discarding the supernatant, a second wash was applied by adding 10 mL of DMEM/F12 medium containing 20% FBS followed by centrifugation at 300 RCF for 3 m. The supernatant was discarded and the cells were resuspended in 1 mL E8-Flex medium supplemented with 10 μM of ROCKi. A cell count was performed and the volume of medium was adjusted to obtain cell density of 500,000 cells/mL. A volume of 1 mL was added to a Matrigel-coated well of a 6-well-plate. Immediately after, 1 mL of the 2× MagC solution was added and gently mixed with the cells using a 1000 μL pipette. The same protocol was followed for the control cells, except the final 1 mL medium addition contained no MagCs. The cells were incubated for 24 h (37 °C, 5% $CO_2$) to obtain mhPSCs and control hPSCs.

## Magnetoid protocol

A 24-well AggreWell™400 plate (Stemcell Technologies) was first prepared following the manufacturer's anti-adherence treatment recommendations. Magnetized and control cells were washed, dissociated, and centrifuged following the above protocol. After a cell count, enough volume of dissociated cells was added to 2.5 mL of E8-Flex medium supplemented with 10 μM of ROCKi to obtain a density of 89,100 cells/mL. A similar procedure was followed for the mhPSCs, however, obtaining 1 mL of E8-Flex medium containing 10 μM of ROCKi and a cell density of 900 cells/mL.

Next, the 24-well AggreWell™400 plate was prepared for cell aggregation. First, 1 mL of mhPSCs was added to a well and aggregated immediately at 300 RCF for 3 m. Subsequently, in the same well, 1 mL hPSCs was added dropwise to the liquid-air interface using a 200 μL pipette. In the control well, 1.01 mL of hPSCs was added directly and topped with 0.99 mL of E8-Flex supplemented with 10 μM of ROCKi. The cells were immediately aggregated at 300 RCF for 3 m.

Once aggregated, another standard 24-well plate was used to house a magnet (Supermagnete, N45, 15 × 15 × 15 mm) with the magnetization axis pointing vertically. The magnet was placed inside one of the wells and covered by the well-plate cover. The well was chosen such that it matched the position the well containing the mhPSCs in the 24-well AggreWell™400 plate. Next, starting from a vertical position of more than 20 cm directly above the 24-well-plate with the magnet, the AggreWell™ was lowered until it was resting securely above the well-plate containing the magnet. The assembly was then moved to the incubator and the cells were incubated (37 °C, 5% $CO_2$) for 24 h.

After the incubation period, the assembly was gently moved out of the incubator and the AggreWell™ was lifted in a vertical manner until it was approximately 20 cm away from the well-plate containing the magnet. This ensured the magnetic aggregates were not magnetically disturbed. 1 mL of media was gently taken out. The magnetic and nonmagnetic aggregates were then separately suspended in the remaining 1 mL of medium using a 1 mL pipette (a wide bore tip should be used to avoid damaging the aggregates). The aggregates were then moved to separate 1.5 mL Eppendorf tubes and centrifuged at 150 RCF for 3 m. The supernatant was removed and 12 μL of differentiation medium was added to each aggregate condition.

A 100 μL 2 kPa Polyethylene glycol (PEG) hydrogel was prepared as previously described[16] for each condition. The 10 μL cell suspension[16] was replaced by a 10 μL aggregate suspension. 10 μL of aggregate-containing hydrogels were placed in wells of a 96-well plate. For the magnetically actuated magnetoid condition, the border wells were chosen. For the control and magnetoids without magnetic actuation, a different well plate was chosen. The well plates were

continuously rotated to prevent organoid aggregation and settling until PEG gelation was visually confirmed[16]. An additional 20 m was required to ensure complete gelation at room temperature conditions. Subsequently 200 μL of differentiation medium (see below) was added to each well. A rectangular magnet (Supermagnete, N45, 30 × 30 × 15 mm) was placed next to the well plate containing the magnetoid for actuation. The magnet was placed on a 1.5 mm plastic spacer so that it was not directly touching the incubator shelf. The well-plate containing the control and non-actuated magnetoids was placed at least 20 cm away from the magnet to reduce magnetic disturbance.

### Human neural tube differentiation magnetization protocol

Organoids were treated immediately after PEG embedding for 3 days with neural differentiation medium comprised of a 1:1 mixture of neurobasal medium (GIBCO) and DMEM/F12 (GIBCO), 1% N2 (GIBCO), 2% B-27 (GIBCO), 1 mM sodium pyruvate MEM (GIBCO), 1 mM glutamax (GIBCO), 1 mM non-essential amino acids (GIBCO) and 2% Penicillin Streptomycin (GIBCO) and supplemented with 10 μM of ROCKi. To induce floor plate identity, organoids were treated from days 3–5 with ROCKi-free neural differentiation medium supplemented with retinoic acid (RA) (Stemcell Technologies) at 0.25 μM and smoothened agonist (SAG) (Stemcell Technologies) at 1 μM. The organoids were subsequently treated with ROCKi-free neural differentiation medium until end point day 11 and the medium was fully refreshed every 2 days. The magnetic field was applied for the entire duration of the 11 day differentiation protocol for all hNTO patterning experiments, except those reported in Supplementary Fig. 9. In Supplementary Fig. 9, to observe the effect of magnetic field removal on pattern stability, we removed the magnet at day 9 and observed FOXA2 expressions at day 11. In a separate experiment, we removed the magnet at day 11 and observed FOXA2 expressions at day 13. In both experiments, the organoids where not subjected to a localized force generated by the MagCs for a duration of 2 days. In these experiments, the differentiation timeline of the unactuated hNTmOs and control organoids was changed accordingly to reflect the same endpoints as the actuated hNTmOs.

### Magnetic nanoparticle toxicity study

Fluorescently labeled MagCs were prepared following the MagC labeling protocol to obtain 2× concentration solutions of 20, 200, 2000, and 10,000 μg/mL. The respective fluorescent particle concentration for each condition was adjusted to maintain their ratio to MNPs. Single hPSCs were plated in wells of a 6-well-plate at a density of 100,000 cells/mL in a volume of 1 mL of E8-Flex supplemented with 10 μM ROCKi. Next 1 mL of the 2× MagC solutions was added to the respective wells and mixed gently with a 1000 μL pipette. The cells were incubated for 24 h, at which point the medium was replaced with ROCKi-free E8-Flex medium and incubated for an additional 48 h. Media change was done carefully to not disturb the MagCs in the wells. Cells were then washed, dissociated, centrifuged and resuspended in 1 mL of E8-Flex medium supplement with 10 μM of ROCKi. A cell count was performed and the total amount of cells per condition was evaluated and normalized to that of the control condition. These values were reported to compare the toxicity of MagCs on hPCS. In separate wells with intact colonies, Calcien AM (Thermofisher) was used to visualize live cells in control and magnetized conditions. The cells were then fixed in paraformaldehyde (PFA 4%, Sigma-Aldrich) for 2 h and gently washed with PBS. The cells were then treated with a permeabilization and blocking solution comprised of 0.3% Triton X (PanREAC AppliChem) and 0.5% BSA (Sigma–Aldrich) in PBS for 30 m. Hoechst was then used to visualize DNA and applied at a concentration of 1:2000 in permeabilization and blocking solution for 2 h at 4 °C followed by 3 PBS washes. Representative images were then obtained using an inverted microscope (Zeiss Axio Observer Z1; Carl Zeiss MicroImaging) equipped with a Colibri LED light source and a 10× air objective.

### Magnetic cluster localization

ZO1 hPSCs were magnetized using a 1 mg/mL MagC solution following the magnetization protocol. The cells were then imaged using confocal microscopy (Leica SP8 DIVE, Leica Microsystems). 3-D reconstructed images were generated from image stacks to demonstrate the localization of MagCs and ZO1 (apical).

### Magnetic cluster size analysis

MagCs were prepared and labeled as described in the above protocol to yield a 2 mg/mL 2×-2 mL solution. A volume of 5 μL was taken and added to 1 mL of DMEM medium and sonicated for 3 m. A 100 μL droplet of the mixture was taken and added to a glass slide. The droplet was imaged using an inverted microscope (Zeiss Axio Observer Z1; Carl Zeiss MicroImaging) equipped with a Colibri LED light source and a 40× air objective. MagCs sizes were assessed using the particle analysis function in ImageJ.

### Magnetic cluster position analysis

1% mhPSC-hNTmO at day 11 were analyzed for the position of their respective MagCs. To obtain a dimensionless analysis for the position of the MagC, for every magnetoids the brightfield image was scaled and projected onto a circular hypothetical organoid while retaining the magnet direction. The MagC shape was then segmented, its position retained and transformed into a binary grayscale image where the MagC shape was attributed a gray scale value of 10 (ImageJ). All MagC images were then projected onto the same space and the intensities were summed. Intensity variation was then color-coded for position density were higher intensities reveal regions most likely to be occupied by MagCs.

### Magnetization efficiency

Magnetized hPSCs were prepared with final MagC concentrations of 10, 100, and 1000 μm/mL following the magnetization protocol. After 24 h incubation, the cells were washed, dissociated, centrifuged and resuspended in 1 mL of E8-Flex medium supplement with 10 μM ROCKi. A 100 μL droplet of the mixture was taken and added to a glass slide. The glass slide was placed under an inverted microscope (EVOS, Invitrogen) and visualized using a ×4 objective. An image was acquired showing the cell positions as they settled towards the glass slide. A magnet (Supermagnete, N45, 15 × 15 × 15 mm) was placed at a distance not greater than 1 mm from the droplet edge. After 2 m, a second image was taken of the same droplet and position. Using ImageJ, a visual comparison determined the percentage of cells that underwent magnetophoresis. Two other conditions were also assessed, 1) control unmagnetized hPSCs, and 2) hPSCs that had been instantly mixed with a final MagC concentration of 1000 μg/mL, following magnetization protocol.

### Magnetic simulations (FEMM) and force estimation

The program Finite Element Method Magnetics (FEMM, version 4.2) was used to conduct two-dimensional axisymmetric magnetics simulations to compute the magnetic field produced by the various magnets used in the study. First, the profile area of the used magnet was constructed and assigned a NdFeB N45 material property from the built-in FEMM material library. The well plate, the PEG hydrogel and the organoid tissue were all considered permeable to the magnetic field and thus ignored in the simulation. All unfilled spaces around the magnet was assigned the material properties of air from the built-in FEMM material library. Standard meshing was chosen prior to running the simulation.

To evaluate the magnetic force produced by MagCs in magnetoids, we first evaluated the field strength within the first well of a 96 well plate and where magnetoids are to be expected. We found the magnetic field produced by the magnet to vary between 0.2 and 0.1 T within the well, which is a range below the magnetic field strength

required for $Fe_3O_4$ nanoparticle saturation magnetization. The magnetic field strength is therefore considered weak and the magnetic moment linearly varies with the applied magnetic field[37]. To calculate the force, a straight line was trace from the corner of the magnet and extended outwards into the air environment. The line length spanned the entire length of a well-plate and the magnetic field values, **B**, were extracted in Tesla. To estimate the magnetic force, we first evaluated the gradient of the squared field, $\partial|\mathbf{B}|^2/\partial y$, along the length of the line section. The magnetic force density, $\mathbf{f}_m = (\chi_{Fe_3O_4}/2\mu_O)\partial|\mathbf{B}|^2/\partial y$, was then calculated where the magnetic volume susceptibility of $Fe_3O_4$ was assumed for MNPs as $\chi_{Fe_3O_4} = 9448.8$ and the permeability of free space $\mu_O = 4\pi \times 10^{-7}$. Each well directly facing the magnet (e.g. six adjacent wells that form a column on a 96-well-plate) was divided into six horizontal tranches with equal thickness. Each tranche was assigned one $\mathbf{f}_m$ value using the simulation data and based on the distance separating the centroid of the tranche to the magnet surface. For any magnetoid in a tranche, the magnetic force $\mathbf{F}_m = \mathbf{f}_m V_{Fe_3O_4}$, where $V_{Fe_3O_4}$ was the volume of the MagC. In this way, the force generated by any MagC in any magnetoid located at any distance away from the magnet surface could be estimated. To estimate the MagC volume, we tested three volume estimation methods 1) cylindrical, 2) elliptical and 3) pixel-depth. In the cylindrical method, the MagC outline is firs traced and the long, a, and short, b, axis of the best fitting ellipse are evaluated. The volume was then evaluated a $V_{cylindrical} = V = \pi a(b/2)^2$. The elliptical volume followed the same procedure as the cylindrical method but where $V_{elliptical} = V = (4/3)\pi ab^2$. In the pixel-depth method, the MagC outline was traced, and the corresponding area, A, was evaluated. Next, the widths at the beginning (evaluated at one quarter the length of the long-axis away from the MagC end), middle and end (evaluated at one quarter the length of the long-axis away from the other MagC end) of the MagC were evaluated and averaged, $w_{avg}$. The MagC volume was then evaluated as $V_{pixel-depth} = A w_{avg}$. Although elliptical geometries are supported by theory, we observed that the MagCs do not strictly adhere this shape as they appear more cylindrical with tapered ends. This may be attributed to the assembly of smaller MagC ellipses to form larger structures. We therefore consider this elliptical method to be an underestimation. Indeed, when comparing the different volume estimation methods we found that the cylindrical and pixel-depth methods resulted in similar volume values with a pixel-depth to cylindrical volume ratio on average ~0.93. By contrast, the elliptical to cylindrical volume ratio on average ~0.67. From visual assessment of MagC shapes, and knowing that two independent methods results in similar volume estimation values, in this study we considered the cylindrical volume estimation method as it required less input parameters that the pixel-depth method. For magnetoids where multiple MagCs appear, such as in the case of 5% and 25% mhPSC-magnetoids, the volume for each MagC is estimated then summed to obtain a total MagC volume per magnetoid. The force is then estimated based on this summed volume, i.e. the obtained force values are that of the cumulative forces from each of the MagCs in the magnetoid. In the 1% and 0.5% mhPSC-magnetoid cases, the majority of magnetoids had one MagC.

### Magnetic force and single-cell fate correlation
The magnetic force was measured as detailed above. Next the distance that separates the centroid of FOXA2+ or PAX6+ nuclei and that of the MagC in the was measured. Both the distance and force were then reported in a correlation plot (Supplementary Fig. 4). Only distances toward the biased zones for each fate (FOXA2−towards the magnet, PAX6−away from the magnet) were reported.

### Magnetic force analysis on single cells
Magnetized hPSCs were prepared with a final MagC concentration of 1,000 µm/mL following the magnetization protocol. After 24 h incubation, the cells were washed, dissociated, counted, centrifuged, and resuspended in E8-Flex medium supplement with 10 µM ROCKi with a volume to obtain $\sim 10 - 100 \text{cell}/\mu L$. This low concentration reduced cell-cell flow perturbation. A glass slide was prepared where a magnet (Supermagnete, N45, $3 \times 3 \times 3$ mm) was glued in place with the magnetization axis aligned with the slide's long axis. The glass slide was then mounted onto the stage of an inverted microscope (Zeiss Axio Observer Z1; Carl Zeiss MicroImaging, 5× objective). Next, 100 µL of the cell suspension was placed onto a glass slide. As the cells underwent magnetophoresis, they began to move through the medium toward the magnet surface and a video (5 frames per second) using the brightfield channel was taken. Using ImageJ, the distance (measured between forward cell edges in consecutive frames) traveled for each cell between frames was measured and divided by 0.2 s to obtain a velocity **u** (~100 µm/s). For a cell radius R of ~10 µm, this velocity resulted in a Reynold's number ≪1, suggesting Stoke's flow conditions where $\mathbf{F}_D = \mathbf{F}_M$ where $\mathbf{F}_D$ is the drag force and $\mathbf{F}_M$ is the magnetic force acting on the MagCs. For each cell, $\mathbf{F}_D = 6\pi\mu R\mathbf{u}$, where $\mu = 8.9 \cdot 10^{-4}$ Pa s is the dynamic viscosity of water, R is the cell radius assuming a spherical geometry, **u** is the calculated cell velocity between frames. The magnetic field produced by the $3 \times 3 \times 3$ mm magnet was then simulated and a force density plot was generated. Knowing $\mathbf{F}_M$ and using the force density plot we could estimate the volume of MagCs attached the cells ~$7.3 \times 10^3$ µm³. When comparing this value to those measured for MagCs in magnetpoid, we saw a difference, suggesting that in addition to the assumed ~1 mhPSCs that is integrated per magnetoid for the 1% mhPSC condition, other unbound MagCs can also find their way into the cellular construct during the aggregation step. It is therefore more accurate, for the purpose of evaluating forces in magnetoids, to estimate the MagC volume in magnetoids directly from images and not simply assume a volume based on the single cell $\mathbf{F}_M$ measurements. These single-cell magnetophresis experiments inferred proper attachment of the MagCs to the cells.

### Short-term magnetoid actuation
A short-term magnetic actuation was conducted on magnetoids embedded in 2 kPa PEG hydrogels, prepared as discussed above and plated in a 96 well plate. Day 11 hNTmOs were observed using an inverted microscope (EVOS, Invitrogen). A magnet (Supermagnete, N45, $15 \times 15 \times 15$ mm) was placed at a ~45° angle (directly next to the adjacent well) with the magnetization axis perpendicular to the plate edge. This setup still allowed MagCs to reorient toward the magnet. An image was then taken of the magnetoid every 5 s for 30 s. Using ImageJ, the displacement of the tissue around the MagC as it reorients towards the magnet was traced and measured by choosing a visible and trackable tissue feature on the brightfield channel. We measured the displacements caused by whole magnetoid rotation and subtracted these from the tissue displacements. The field of view was divided into an array of tissue sections and the displacements, $d_{tissue}$, in each tissue section were normalized to the displacement of the part of the MagC, $d_M$, in that row to provide a dimensionless displacement value d*. The distance, $x$, from the centroid of each tissue section to the edge of the MagC in the same row was normalized to the displacement of the part of the MagC in that row to provide a dimensionless coordinate value c*. A d*-c* correlation and heatmap was then reported.

### Long-term magnetoid actuation
A similar setup as the one discussed above was used. However, here the magnet was placed for longer durations of 5, 30, 60, and 120 m under incubation conditions of previously magnetically actuated day 11 hNTmOs. At time $t_0$ without the magnet placement, the reference angle of the MagC was imaged. At every subsequent time point, the magnet was taken away and the new MagC angle was imaged. The recoil angle difference, α, was calculated by subtracting the reference angle (original position) from the new angle (final position) and was then reported.

## Growth and proliferation studies

To determine the size of hNTmOs, day 11 magnetoids were imaged using an inverted microscope (EVOS, Invitrogen). Using ImageJ, the projected area of organoids was traced and converted to an equivalent diameter assuming a circular area.

To determine the biased growth of magnetoids when compared to control or unactuated magnetoids, all organoids brightfield images were taken at days 1 and 11. A contour line was traced to mark the organoid boundary at days 1 and 11 and the images were superimposed using non-mobile reference points in the surrounding matrix as anchor points. Next, a line section, $P_m$, was traced connecting the closest point to the magnet surface of the contours at days 1 and 11. A similar line section, $P_a$, was traced to connect the furthest point to the magnet at both days. The ratio of the growth towards the magnet, $P_m$, and growth away from the magnet, $P_a$, was calculated where a value <1 and >1 indicated biased growth away and toward the magnet respectively while a value ~1 represented an unbiased growth direction.

To determine the proliferation bias, day 11 hNTOs and hNTmOs were fixed and stained with EdU and images were taken using an inverted microscope (Zeiss Axio Observer Z1; Carl Zeiss MicroImaging, 10× objective). For all cases, a line segment dividing the organoid into two halves and passing through the organoid centroid was traced. The average EdU intensity in the halves facing the magnet, $EdU_m$, were normalized to that of the respective halves facing away from the magnet, $EdU_a$, where a value <1 and >1 indicated biased proliferation away and toward the magnet respectively while a value ~1 represented an unbiased proliferation direction.

To investigate the cell density bias, the averaged Hoechst signal from organoid halves from day 11 hNTOs and hNTmOs was analyzed in the same way as the EdU analysis. A Hoechst intensity ratio <1 indicated less cells in the organoid half facing the magnetic and a value >1 indicated less cell density in the organoid half facing away from the magnet, while a value of ~1 represented a balanced cell density between the halves of the organoid.

## Fate specification and patterning studies

The induction and patterning efficiencies of FOXA2, NKX2.2, OLIG2, NKX6.1, ISL1/2, and PAX6 were visually inspected across the various conditions using an inverted microscope (Zeiss Axio Observer Z1; Carl Zeiss MicroImaging, 10× objective). First organoids were fixed with 4% PFA for 2 h followed by PBS washing and then permeabilization and blocking using a solution of 0.3% Triton X (PanREAC AppliChem) and 0.5% BSA (Sigma-Aldrich) for 30 min. Next, primary antibodies suspended in blocking and permeabilization solution were applied to the samples for 24 h. The following primary antibodies were used in this study, FOXA2 (Santacruz (sc-374376), mouse monoclonal, dilution 1:200), NKX2.2 (DSHB (DSHB − 74.5A5), mouse monoclonal, dilution 1:200), OLIG2 (R&D systems (AF 2418), goat polyclonal, dilution 1:200), NKX6.1 (DSHB (DSHB - F55A10), mouse monoclonal, dilution 1:200), ISL1/2 (DSHB (DSHB − 39.4D5), mouse monoclonal, dilution 1:200), TUBB3 (Biolegend (802001), rabbit polyclonal, dilution 1:200), PAX6 (Biolegend (901301), rabbit polyclonal, dilution, 1:200). The samples were subsequently washed three times using PBS over a period of 24 h. Immunolabeling was then performed using secondary antibodies suspended in blocking and permeabilization solution for an additional 24 h. The following secondary antibodies were used in this study, Alexa Fluor 555 (Invitrogen (A-31570), donkey polyclonal, donkey anti mouse, dilution 1:500), Alexa Fluor 647 (Invitrogen (A-31573), donkey polyclonal, donkey anti rabbit, dilution 1:500), Alexa Fluor 647 (Invitrogen (A-21447), donkey polyclonal, donkey anti goat, dilution 1:500). The samples were subsequently washed three times using PBS over a period of 24 h before imaging.

Pattering efficiencies of FOXA2, NKX2.2, OLIG2, NKX6.1, ISL1/2, and PAX6 in organoids were assessed as previously reported[16]. Briefly, fluorescent images were assessed using ImageJ to obtain the area of the expression region of the fate as well as the average intensity of the expression in that area. Next, the projected area of the entire organoid or magnetoid was evaluated and an average intensity of the fate expression was obtained. We next evaluated the area ratio (AR) where the fate expression area was divided by the organoid area. We also evaluated the intensity ratio (IR) where the average fate intensity of the expression region was divided by that of the organoid or magnetoid area. The fate was considered as scattered for (1) AR > 0.5, and (2) AR < 0.5 but IR < 20%. By contrast, the fate expression was found to be pattered for AR < 0.5 and IR > 20%. We note that one of the biological replicates in the unactuated hNTmO cases (Fig. 5a) yielded no patterning of PAX6 expressions. We believe this to be an outlier as all other 3 biological replicates cluster together.

To evaluate the percentage of dorsoventral patterning we assessed the expression of NKX6.1 (ventral fates) and PAX6 (intermediate fates) in the same organoid and reported the overall fraction of double positive organoids.

To interrogate expression phenotypes for each of the markers, the marker expression image was scaled and projected onto a circular hypothetical organoid while retaining the original orientation. For each marker, the hypothetical organoid images were projected onto the same space and the intensities were integrated. Intensity variation was then color-coded. Higher intensities revealed regions with highest marker positional bias in the organoid space across 50 to 100 organoids from 3 to 5 biological replicates. To quantify the marker directional bias upon internal force generation, inverted microscopy images were assessed, in a similar manner as for EdU stains, for each of the markers separately.

To investigate pattern organization with reference to the magnetic field direction, we evaluate the projected intensity profiles of FOXA2, NKX6.1, NKX2.2, OLIG2 and PAX6 of the integrated marker expression every 30° from −90° to 90° from the original organoid orientation. The profiles were first normalized then compared to an ideal square intensity pattern and a correlation was established using the corrcoef function MATLAB (R2018a, The MathWorks Ink.). A correlation heatmap was generated using the R (version 4.1.0) package heatmap.2 using RStudio (RStudio Team (2020). RStudio: Integrated Development for R. RStudio, Inc., Boston, MA URL http://www.rstudio.com/). To visualize the enhanced pattern organization in magnetoids we merged the top 20% of the integrated expressions of FOXA2, NKX6.1, OLIG2, and PAX6 fates and reported their profiles at the original organoid orientation.

To evaluate the magnetic field effect, MagC-free hNTOs were subjected to a magnetic field similar to the hNTmO case. The cytoarchitecture and FOXA2 and PAX6 induction rates were then evaluated.

## Quantification and statistical analysis

Two-way ANOVA statistical tests and unpaired two-tailed $t$ test with corrections were used where appropriate with a 95% confidence interval (GraphPad Prism 6, Version 6.01, GraphPad Software, Inc.). Statistical significance was considered for all comparisons with $p < 0.05$. Microsoft® Excel® for Microsoft 365 MSO (Version 2212) was used for fate induction and patterning quantifications.

## Reporting summary

Further information on research design is available in the Nature Portfolio Reporting Summary linked to this article.

## Data availability

All data supporting the findings of this study are available within the article and its supplementary files. Any additional requests for information can be directed to, and will be fulfilled by, the lead contact. Source data are provided with this paper.

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

## Acknowledgements

This work was supported by the FWO grants G087018N, G086622N and FWO postdoctoral fellowship 1217220N, KU Leuven grants C14/17/111, C32/17/027, IDN/20/007, IDN/22/012 and by an Allen Distinguished Investigator Award (AR), a Paul G. Allen Frontiers advised grant of the Paul G. Allen Family Foundation.

## Author contributions

A.A. designed all experiments and performed experiments and analysis. N.K. conducted nanoparticle embedding experiments and performed induction and patterning experiments and analysis. K.V.D. performed magnetic simulations, magnetic force analysis, and cytoskeleton assessment of magnetoids. B.D. assisted with microscopy. G.R. assisted with protocol establishment. A.A., N.K., K.V.D., and A.R. interpreted the data and edited the manuscript. A.A. and A.R. wrote the manuscript.

## Competing interests

The authors declare no competing interests.
