## [Peer Review File · Nature Communications]

REVIEWER COMMENTS

Reviewer #1 (Remarks to the Author):

The work “Targeted mechanical stimulation via magnetic nanoparticles guides in-vitro tissue development” introduces a method for in situ mechanical stimulation of organoids using embedded magnetic nanoparticles, termed “magnetoids”, for mechanical actuation and further investigation of local mechanotransduction during development. Some clarifications may be needed before publication is recommended.

- Although the magnetic labeling of the cells serves some initial magnetic characterization and fluorescence visualization purposes, the authors claim that the magnetic clusters do not undergo endocytosis, and that after magnetic field application they leave the cells to self-assemble into a rod shape. What is then the ultimate purpose of the first hPSCs’ 24h incubation with these clusters? Could the same final rod-shape result be obtained by simply adding the clusters to the cell mix at different concentrations and then applying a magnetic field? Would there be differences in the resulting applicable forces?

- Considering that magnetic clusters leave their cell chaperons, and the changes in cell arrangement within the magnetoids through maturation, it can be misleading to reference the percentage of magnetized cells. It might be better to reference this throughout the manuscript as an initial proportion. An alternative would be to make reference to the size/volume of the resulting magnetic rod in comparison to the magnetoid’s.

- The recoil angle definition is confusing in Fig. 3. It is defined in the text as the recovery angle of the rod towards the initial position upon removal of the magnetic field (Fig. 3d, third panel). However, in the fourth panel of Fig. 3d, it is shifted to be the angle of actuation. This should be clarified. Are the faint lines in the magnetoid images meant to represent the initial and final actuation rod positions?

The text page 7 should be changed according to the right definition of the recoil angle: If longer magnetic field application times trigger cellular reorganization that prevents the cluster to return to its original position, the recoil angle should be smaller as the incubation time increases. Figure 3e shows the contrary.

- To the best of my reading, the duration of the magnet application for the patterning analysis is missing in the main text. Was this maintained during the 11 days of growth? It would be of interest to show if the patterning is maintained upon removal of the magnet in order to better characterize the magnetoid tissue development as a response to biomechanical cues.

- Updating the references to more recent ones would be beneficial. Only a couple are from after 2020, and mostly self-citations. For instance, Ref 21 could be replaced by *Advanced Functional Materials* 2020, 30 (25), 2002541.

Minors:

- Abstract: "in these organoid" -> organoids
- Introduction: The sentence "The ability to tailor..." must be reworked. It can't be understood.
- Typo in Figure 3d, where both instances of "angle" should read "angle".
- Supplementary Fig.4: Should be from Fig. 4b or 4a.
- Supplementary Fig.7: Should be Fig. 5c

Reviewer #2 (Remarks to the Author):

In this work from Fattah et al., a tissue fabrication strategy for generating organoids with incorporated magnetic particles is presented, and the resulting tissues termed "magnetoids" are subjected to forces via placement proximal to a permanent magnet. Presented results convincingly demonstrate that magnetically responsive inclusions influence the localization of proliferative cells and spatially bias the presence of PAX6 vs. FOXA2 expressing cells. Given recent interest in controlling the generally heterogeneous and spontaneous cell specification that occurs in typical organoid models, this reviewer views the work as interesting from a conceptual perspective. However, major criticisms are the relatively minor effect shown from applying magnetic forces (altered proliferation and a modest bias in gene expression) and lacking insight on how the applied forces are lead to the observed differences. The first three figures are largely confirmatory to what is already known about associating magnetic particles with cells, eg. magnetic particle-loaded cells can be moved in suspension which is the basis for magnetic assisted cell sorting. Figure 4 and 5 are more interesting, but not entirely exciting or insightful. A more pronounced effect of applying forces to magnetoids in terms of specification of distinct tissues (rather than simply gene expression) or deeper insight into whether the observed differences arise from tensile vs. compressive forces, or alternatively simply densification of cells proximal to the magnet, would strengthen this work.

Reviewer #3 (Remarks to the Author):

This paper showed local force by magnetic approach affects neural tube organoid development in vitro. The idea to generate magnetic clusters in organoids is novel and interesting, however there are some concerns as follows;

Major points

1. The authors showed magnetoid generation (fig1), the actuation of magnetic force in organoid (figs 2-3), the local actuation affects growth and proliferation in organoids (fig4), and the local actuation induce tissue patterning of floor plate/intermediate zone (fig5). It looks the logical flow of fig1-4 is smooth, but there still exist some gaps between figs1-4 and fig5. Why biased growth/proliferation induce regional patterning? How magnetic forces trigger the patterning? What types of signal cascade work during the magnetic actuations. These questions are still exist, and authors can fill the gaps between fig4 and 5 by answering these questions, and make clear the correlation between each figure.

2. Relating to point 1, the data in fig5 need more detailed analysis. The authors showed floor plate and intermediate domains in neural tube organoids but the expression of ventral domain markers can be added for better interpretation of the data. There are some gap between expression of FOXA2 and PAX6 in fig5c, and there will exist ventral domain.

Besides, the authors showed FOXA2 and PAX6 expression separately, but it is better to show in the same samples.

3. The expression of PAX6 decrease in fig5 b No mag, but there is not interpretation about this data. Besides, the staining of PAX6 in supplementary fig6 is not clear.

Minor points

1. Is it possible to explain more detail about how MagCs forms rod shaped structure.

2. I have concerns about the naming of "magnetoid". The term "organoid" means organ-like tissues, then "magnetoid" means magnet like tissue and it looks the naming does not reflect the data.

3. The explanation of supplementary fig7 can be changed for better readability. The correlation of magnetic log and nucleus-MagC distance is difficult to understand for me. Please add more explanation in figure legend or text.

4. There are some overstatements.

Abstract line12~: "This versatile approach, ~~"

Authors showed neural tube patterning, and the descriptions can be focused on the authors data.

Page8 line25~: "suggesting that bending of the ~~~~"

This interpretation need in vivo corresponding data.

Page10 line21~: "Moreover, while we demonstrate ~~~~"

Authors showed neural tube patterning, and the descriptions can be focused on the authors data.

Fattah et al. Response to reviewers

We thank the reviewers for their interest in our manuscript and for their detailed comments. Below we provide a reply to these comments, and include an updated manuscript where additional text is highlighted in yellow. Please note that part of the response to Reviewers 2 and 3 overlap and for clarity we have duplicated our response accordingly.

Reviewer 1

- 1) Although the magnetic labeling of the cells serves some initial magnetic characterization and fluorescence visualization purposes, the authors claim that the magnetic clusters do not undergo endocytosis, and that after magnetic field application they leave the cells to self-assemble into a rod shape. What is then the ultimate purpose of the first hPSCs' 24h incubation with these clusters? Could the same final rod-shape result be obtained by simply adding the clusters to the cell mix at different concentrations and then applying a magnetic field? Would there be differences in the resulting applicable forces?

Instantly mixing the magnetic nanoparticles with the cell populations did not result in proper attachment to the cells as shown in Fig.1c of the manuscript. When aggregating this instant mix approach in the aggregewells, it resulted in the formation of multiple magnetic clusters within the organoid, as a result, free-floating nanoparticles in the medium during aggregation (see additional Figure below). While the total applied force will not differ much compared to that applied from a single rod with the same volume of MagCs, the applied force in this case will be distributed over multiple points (one for each cluster position), resulting in an unwanted added degree of randomness to the system. We have thus concluded that to reproducibly produce magnetoids with single rod magnetic clusters an incubation step with the cells is required. This helps form cell-cluster attachments and allows cells to chaperone the nanoparticles during the first aggregation step of the sequential aggregation to the bottom-most region of the magnetoid, reducing the proximity between the MagCs and allowing them to cluster into a single rod under a magnetic field.

To summarize, our experiments indicate that the same final rod-shape result could not be obtained by adding the clusters to the cell mix and then applying a magnetic field. This would result in multiple smaller rods and a more heterogeneous applied force.

Figure for Reviewer 1 – Comment 1: Instant mixing of magnetic nanoparticles with the cells during the aggregation step (Instant mix) results in multiple MagC clusters in the magnetoids. Preincubation of magnetic nanoparticles with the cells and performing a sequential aggregation (24 h preincubation + sequential aggregation) results in single rod in the magnetoids. Scalebar 50 μm .

To make it clear to the reader we have modified the manuscript; below is an excerpt from the “Magnetoid generation” section.

“Magnetized pluripotent aggregates with localized MagCs (**Fig.1a**) were produced by first centrifugating mhPSCs followed by a second centrifugation step to aggregate the non-magnetic hPSCs. This consecutive centrifugation allows mhPSCs and their MagC tags to collect together at the bottom of the aggregation well. We subsequently perform a 1 day incubation under a magnetic field. Because the MagCs are only transiently attached to their cellular chaperones, once they become magnetized under the magnetic field, a force is generated within the MagCs and they leave the cell surfaces in an attempt to travel towards the magnet. However, because MagCs are in close proximity to each other, they undergo directed assembly as they interact and influence each other through their own induced magnetic fields, allowing them to build rod-shaped MagCs inside the organoids which ensures force localization. Instantly mixing magnetic nanoparticles with non-magnetized hPSCs produced magnetoids with multiple MagC clusters and thus force foci, an undesirable effect that diffuses the force rather than localizes it. These magnetized pluripotent aggregates were then embedded in PEG hydrogels with a stiffness of 2 kPa as previously described¹⁶, and could then be differentiated under constant or periodic magnetic actuation to impose internal local forces through the displacement of the rod-shaped MagCs.”

- 2) Considering that magnetic clusters leave their cell chaperons, and the changes in cell arrangement within the magnetoids through maturation, it can be misleading to reference the percentage of magnetized cells. It might be better to reference this throughout the manuscript as an initial proportion. An alternative would be to make reference to the size/volume of the resulting magnetic rod in comparison to the magnetoid's.

As advised by the Reviewer, we revised the manuscript as well as **Figure 2** to described the percentage of magnetized cells as “initial proportion” or “initial percentage”.

- 3) The recoil angle definition is confusing in Fig. 3. It is defined in the text as the recovery angle of the rod towards the initial position upon removal of the magnetic field (Fig. 3d, third panel). However, in the fourth panel of Fig. 3d, it is shifted to be the angle of actuation. This should be clarified. Are the faint lines in the magnetoid images meant to represent the initial and final actuation rod positions?

To clarify, the faint black, blue and black ovals in **Figure 3d** are meant to represent the original position, final position and the position under magnetic field respectively. We agree with the reviewer that this could be shown and stated more clearly and as a result, we modified the manuscript in the “**Short and long term magnetic actuation in magnetoids**” section to clarify the new definition of α as the recoil angle difference:

“In order to investigate the transition from fast (bursts) to slow (constant) forces, we evaluated the MagCs recoil angle difference, α , which is the angular shift between the original (**Fig.3d** faint black oval) and final (**Fig.3d** blue oval) MagC position upon recoil from the magnetized position (**Fig.3d** black oval) after removal of the magnetic field (**Fig.3d**).”

We have also revised **Figure 3d**:

Figure 3d: Schematic representation of MagC actuation and recoil (faint black oval: MagC initial position, black oval: MagC position under a magnetic field, blue oval: MagC final position after removal of magnetic field). Representative images of MagC actuation and recoil during long-term magnetic incubation. (n = 4 independent experiments for 35 hNTmOs). Scalebar 50 μ m.

- 4) The text page 7 should be changed according to the right definition of the recoil angle: If longer magnetic field application times trigger cellular reorganization that prevents the cluster to return to its original position, the recoil angle should be smaller as the incubation time increases. Figure 3e shows the contrary.

The reviewer is correct that the recoil angle from the MagC position under the magnetic field will be smaller as the incubation time increases, however, after our clarification in the above comment, the reported angle is in fact the recoil angle difference, which is the angle between the initial and final positions. By this definition, longer magnetic field application times will result in larger angles between initial and final positions because of the decrease in recoil angle.

- 5) To the best of my reading, the duration of the magnet application for the patterning analysis is missing in the main text. Was this maintained during the 11 days of growth? It would be of interest to show if the patterning is maintained upon removal of the magnet in order to better characterize the magnetoid tissue development as a response to biomechanical cues.

The magnetic field is applied for the full 11 days of differentiation. To clarify this point we have modified the **“Local magnetic actuation guides tissue-wide patterning in human neural tube magnetoids”** section, below is an excerpt from the manuscript.

“Magnetoids were differentiated towards the neural fate upon embedding in the PEG hydrogels, using retinoic acid (RA) and smoothed agonist (SAG) treatment from days 3 to 5 to posteriorize and ventralize emergent cell fates. The magnetic field was applied to the magnetoids for the entire 11 day duration of the experiment.”

To address the reviewer’s interesting question regarding the effect of removal of the magnet, we performed two additional sets of experiments. To address whether the localized force is necessary to maintain patterning after it is established, we removed the magnetic field (1) at day 9 and assessed FOXA2 expression at day 11, and (2) at day 11 and assessed FOXA2 expression at day 13. The results of these experiments are reported in the new **(Supplementary Fig.9)**. In both cases, organoids where

mechanical stimulation via magnetic fields was removed at late time points retained a high FOXA2+ patterning efficiency similar to those actuated continuously, compared to unactuated controls. These results indicate that removal of the magnetic field does not significantly affect the pattern once it is established, and suggests that FOXA2+ subpopulations form stable self-sustained patterns that are not hindered by the removal of the mechanical stimulation which initiates the pattern formation.

In light of these new observations we have modified the **“Local magnetic actuation guides tissue-wide patterning in human neural tube magnetoids”** section, below is an excerpt from the manuscript.

“To address whether the localized force is necessary to maintain patterning after it is established, we removed the magnetic field (1) at day 9 and assessed FOXA2 expression at day 11, and (2) at day 11 and assessed FOXA2 expression at day 13 (**Supplementary Fig.9**). In both cases, organoids where mechanical stimulation via magnetic fields was removed at late time points retained a high FOXA2+ patterning efficiency (magnetic field removal at day 9 ~57.7%, magnetic field removal at day 11 ~62.7%) similar to those actuated continuously (**Fig.5a**), compared to the respective unactuated controls. These results indicate that removal of the magnetic field does not significantly affect the pattern once it is established, and suggests that FOXA2+ subpopulations form stable self-sustained patterns that are not hindered by the removal of the mechanical stimulation which initiates the pattern formation.”

We have included the new figure below (**Supplementary Fig. 9**) in the revised manuscript.

Supplementary Fig.9: Schematic representation of the experimental timeline. Representative images showing Day 11 and Day 13 organoids with MagC positions and FOXA2 expressions in different conditions (n = 3 independent experiments). Percentages of FOXA2+ organoids and patterning in FOXA2+ organoids in different conditions (n = 3 independent experiments for >70 organoids for all datapoints per condition). Scalebar 50 μ m. Error bars are SEM.

We have also updated the **“Human neural tube differentiation magnetization protocol”** section in the **“Materials and methods”**:

“The magnetic field was applied for the entire duration of the 11 day differentiation protocol for all hNTO patterning experiments, except those reported in Supplementary Fig.9. In Supplementary Fig.9, to observe the effect of magnetic field removal on pattern stability, we removed the magnet at day 9 and observed FOXA2 expressions at day 11. In a separate experiment we removed the magnet at day 11 and observed FOXA2 expressions at day 13. In both experiments, the organoids were not subjected to a localized force generated by the MagCs for a duration of 2 days. In these experiments, the differentiation timeline of the unactuated hNTmOs and control organoids was changed accordingly to reflect the same endpoints as the actuated hNTmOs.”

- 6) Updating the references to more recent ones would be beneficial. Only a couple are from after 2020, and mostly self-citations. For instance, Ref 21 could be replaced by Advanced Functional Materials 2020, 30 (25), 2002541.

We revised the manuscript accordingly to include more up to date references. Specifically, we updated the abovementioned reference, below is an excerpt from the “**Introduction**” section of revised manuscript.

“This type of remote actuation is scalable to multicellular constructs. For example, microfabricated magnetic patterns can impose cyclic mechanical forces on embryonic stem cell aggregates labelled with magnetic nanoparticles, providing better cardiac differentiation and leading to higher efficiency of beating cellular constructs²⁴.”

We also updated the manuscript with additional recent references as suggested by the reviewer.

“Dynamic matrices allow for changes in matrix properties for example using light to induce crosslinking to control axon projections in spinal cord organoids¹⁴ or provide dynamic matrix rearrangement with reversible hydrogen bonding to promote crypt formation in intestinal organoids¹⁵. However, such matrices do not impose active forces on the tissues, thereby missing the recapitulation of dynamic mechanical events seen in-vivo.”

“Optical tweezing can deliver subcellular point forces but is limited to piconewton ranges and thus to applications requiring only weak forces, such as the mechanical stimulation of ion channels¹⁷ and protein complexes related to synaptic activity¹⁸ in neurons”

We also updated the revised manuscript to address the minor comments below.

- 7) Abstract: “in these organoid” -> organoids

This typo has been corrected.

- 8) Introduction: The sentence “The ability to tailor...” must be reworked. It can’t be understood.

This section has been rewritten for clarity:

“Synthetic extracellular matrices such as polyethylene glycol (PEG) hydrogels have shown how matrix stiffness modulation could not only sustain organoid cell growth and morphogenesis¹¹, but could also guide coordinated multicellular differentiation¹² and patterning¹³.”

- 9) Typo in Figure 3d, where both instances of “angle” should read “angle”.

This typo has been corrected.

- 10) Supplementary Fig.4: Should be from Fig. 4b or 4a.

This information has been updated to “**Fig. 4a**”.

- 11) Supplementary Fig.7: Should be Fig. 5

Now Supplementary **Figure 10**. This information has been updated to “**Fig. 5a**”

Reviewer 2

- 1) In this work from Fattah et al., a tissue fabrication strategy for generating organoids with incorporated magnetic particles is presented, and the resulting tissues termed “magnetoids” are subjected to forces via placement proximal to a permanent magnet. Presented results convincingly demonstrate that magnetically responsive inclusions influence the localization of proliferative cells and spatially bias the presence of PAX6 vs. FOXA2 expressing cells. Given recent interest in controlling the generally heterogeneous and spontaneous cell specification that occurs in typical organoid models, this reviewer views the work as interesting from a conceptual perspective. However, major criticisms are the relatively minor effect shown from applying magnetic forces (altered proliferation and a modest bias in gene expression) and lacking insight on how the applied forces are lead to the observed differences. The first three figures are largely confirmatory to what is already known about associating magnetic particles with cells, eg. magnetic particle-loaded cells can be moved in suspension which is the basis for magnetic assisted cell sorting. Figure 4 and 5 are more interesting, but not entirely exciting or insightful. A more pronounced effect of applying forces to magnetoids in terms of specification of distinct tissues (rather than simply gene expression) or deeper insight into whether the observed differences arise from tensile vs. compressive forces, or alternatively simply densification of cells proximal to the magnet, would strengthen this work.

We thank the reviewer for this comment, and would like to respond both in clarifying the novelty of the work with regards to differences in approach in this work compared to previous nanoparticle-based studies, as well as providing results from multiple new experiments focusing on cell packing and including other ventral domains. As a result of the analysis of these new experiments, we believe that we are providing a better understanding and appreciation of the pronounced role of applying localized forces on organoid patterning. Specifically, we can now demonstrate that local magnetic forces provide a new degree of control to specify distinct tissue domains in organoids, provide insight into the role of compressive vs. tensile forces in these observed differences, and a quantification of the role of such localized forces in modulating cell density.

The reviewer is correct that diamagnetophoresis and magnetophoresis of cells are particularly well characterized phenomena used in cell sorting and tissue engineering. Magnetic nanoparticle labeling of cells is most often based on particle internalization by endocytosis or on conjugation to membrane receptors via ligand or other protein labeling of the nanoparticles. Here, we used nonfunctionalized nanoparticles for non-permanent interactions with cells. We demonstrate in our new approach a way to generate cell-nanoparticle interactions which are strong enough to chaperone cells, but at the same time weak enough to allow the nanoparticle to leave their host cells and reorganize into rods inside the organoids. In our study, we also show the viability, effect on cytoskeletal organization, and force modulation effects on organoids, which to the best of our knowledge has not been shown before. We would like to note that more pronounced forces can certainly be achieved using more nanoparticles, however, this resulted in cytoskeletal disorganization (**Figure 2** of the manuscript), which impairs cell fate specification and organoid patterning. For this reason we chose to maximize the forces that retained cytoskeletal organization, i.e. 1% mhPSCs.

We would like to clarify that the reported FOXA2 and PAX6 expressions, established markers of distinct floor plate and intermediate domain tissues respectively, are not observations of gene expression but of protein expression obtained by immunohistochemistry, which confirm protein presence.

We also show that tissue compression in the direction of the magnet and in the proximity of the MagC reduces the proliferative capacity of the cells in that region, while further away regions retain their normal cell cycle. This is demonstrated through EdU staining, showing that the proliferation in hNTmOs subjected to a magnetic field is reduced in the zone facing the magnet (i.e. compressive region) compared to the zone facing away from the magnet (**Figure 4a**) (i.e. tensile region). As suggested by the reviewer, we also performed a cell density assessment using Hoechst

signals averaged over the area to demonstrate the half of hNTmO, containing the compressed and less proliferative regions, on average have less cells (Hoechst signal) when compared to the opposite half (New **supplementary Figure 6**). This analysis indicates that the local compressive forces are sufficient to decrease the cell density in that area (i.e. do not locally densify the tissue but on the contrary lead to reduced cell proliferation and density). We have revised the **“Local magnetic actuation biases growth and proliferation in human neural tube magnetoids”** section, below is an excerpt from the manuscript.

“Zones of reduced proliferation also reduced the cell density in the hNTmO halves that are facing the magnet, as measured by the lower averaged Hoechst signal in these halves, compared to those facing away from the magnet resulting in an average Hoechst intensity ratio (HIR) <1 (**Supplementary Fig.6**). In contrast, in control organoids and hNTmOs not subjected to a magnetic field, cell densities on either half were comparable with HIR ~1. This result suggests that local compressive forces serve to decrease cell proliferation and cell density in these regions.”

We have also included the below figure (**Supplementary figure 6**) in the manuscript.

Supplementary Fig.6: Representative images for Hoechst staining of day 11 hNTmOs under various actuation conditions (n = 4 independent experiments from Figures 4 and 5). Schematic representation of biased cell density in hNTmO under the influence of magnetic field. Hoechst intensity ratio of day 11 hNTmOs for various actuation conditions (n = 4 independent experiments from Figures 4 and 5 for a total of 100 organoids per condition). Statistical analysis was determined by unpaired two-sided t-test Control-Mag $p < 0.0001$, No Mag-Mag $p < 0.0001$. Scalebar 50 μm . Error bars are SEM.

We have updated the **“Growth and proliferation studies”** section in the **“Materials and methods”**, below is an excerpt from the manuscript.

“To investigate the cell density bias, the averaged Hoechst signal from organoid halves from day 11 hNTOs and hNTmOs was analyzed in the same way as the EdU analysis. A Hoechst intensity ratio <1 indicated less cells in the organoid half facing the magnetic and a value >1 indicated less cell density in the organoid half facing away from the magnet, while a value of ~1 represented a balanced cell density between the halves of the organoid.”

To explore whether applying forces to magnetoids would have a pronounced effect in terms of specification of distinct tissues, as requested by the reviewer, we have performed extensive new experiment to include more fates and a new analyses that demonstrate the pattern-guiding capacity of local forces.

To do so, we characterized additional ventral domains NKX2.2 (p3), OLIG2 (pMN), NKX6.1 (p3-pMN-p2) as well as motor neuron markers ISL1 and pan neuron marker TUBB3. The results of these experiments are detailed in the updated **Figure 5**, where we assessed patterning of each of these markers. To gain an overview of the overall patterning morphology and directional bias, for each fate we projected the marker expressions from all observed organoids onto the same space, scaling respective organoid size where necessary to ensure proper mapping while keeping track of the magnet position. This resulted in a heatmap of integrated expression for each fate (**Figure 5b**). The results clearly demonstrate hotspots of fate expression, which convey the most likely regions a fate will be expressed within the organoid. Of interest is the magnetically actuated condition where FOXA2 and NKX6.1 are primarily expressed in the organoid half closest to the magnet (less proliferative and compressed regions), contrasting with the expression of PAX6, which primarily occurs on the opposite half (more proliferative regions facing away from the magnet). Therefore, we show, as suggested by the reviewer, that the FOXA2 and PAX6 domains are mutually exclusive when magnetically actuated. Strikingly, we also note that OLIG2 and ISL1 expressions flank the hNTmO centerline bilaterally, in an expression pattern similar to their in-vivo counterpart. Comparatively, in the control and unactuated hNTmO conditions, although hotspots do occasionally occur, all fates and particularly FOXA2, NKX6.1 and PAX6 form expansive domains that are in sharp contrast to the magnetic stimulation case where domains are spatially defined.

We have quantified these biases in expression towards or away from the magnet side for each marker as we previously did for FOXA2 and PAX6. This was done by measuring the marker intensity in the two halves of the organoids (facing towards and away from the magnet) and reporting the ratio of these two values (**Figure 5c**). We found the most significant difference to be associated with FOXA2, NKX6.1 and PAX6, which are the domains that most polarize towards or away from the magnet.

To further illustrate the pattern direction correlation with force localization, we analyzed the integrated FOXA2, NKX6.1, NKX2.2, OLIG2 and PAX6 expressions in scaled organoids along various projection angles (**Figure 5d**). We chose these transcription factors because they represent ventral domain patterning along the dorsoventral axis. We then compared, at each projection angle, the correlation with an ideal pattern that minimally recapitulates the ventral organization of the in-vivo neural tube. We divided this ideal pattern into 4 domains focusing on the ventral side 1) the floor plate (FOXA2), 2) p3 domain (NKX2.2) 3) progenitor motor neuron domain (OLIG2) and 4) intermediate/dorsal domains (PAX6), which are mutually exclusive, and 5) NKX6.1, which colocalizes with FOXA2, NKX2.2, and OLIG2. Fates in the control and unactuated hNTmO cases occurred largely at random locations, therefore, as expected, rotation of the organoids along 180° did not yield strong correlation with the ideal pattern. In the magnetically actuated hNTmOs, a strong correlation with the ideal pattern was observed at a rotation angle of 0°, which is the unaltered orientation with reference to the magnet position. This analysis demonstrated that pattern organization is a function of magnet position and that this organization is not recapitulated in magnet-free and control conditions at any other angle.

To further illustrate this force-induced organization, we combined the top 20% of the integrated fluorescence intensity of FOXA2, NKX6.1, OLIG2 and PAX6 expressions and reported the analyzed expressions along a rotation angle of 0°. Although this is an integrated expression that is accumulated over many organoids, the results clearly demonstrate the enhanced pattern organization in magnetic stimulation conditions. These results suggest that although not all the fates are recapitulated in each organoid, on average, in magnetic stimulation conditions, fates are restricted to specific regions. Here FOXA2 emerges in the ventral regions (towards the magnet) where it is largely mutually exclusive with OLIG2 expression, which flanks the lateral sides, similar to *in vivo* observations. While NKX6.1 clearly overlaps with FOXA2 and OLIG2 domains, it is mutually exclusive with PAX6

expression, which is restricted to the dorsal-most (away from the magnet) side of the reconstructed organoid. Control and unactuated hNTmOs clearly do not exhibit similar level domain organization, emphasizing the importance of force localization. Local forces can thus be considered as a mechanical frame of reference in organoids to orient and guide patterns.

We have revised the “**Local magnetic actuation guides tissue-wide patterning in human neural tube magnetoids**” section to reflect these new experimental observations and analyses:

“We first investigated the effect of the magnetic field alone on the induction of ventral and intermediate neural tube domains by analysing expression of FOXA2 and PAX6 respectively. When subjecting MagC-free organoids to a magnetic field, we found that cytoarchitecture and frequency of FOXA2 and PAX6 were similar to control conditions not subjected to a magnetic field, indicating that magnetic fields alone do not alter the biological responses in this model system (**Supplementary Fig.7**). We further expanded our analysis to assess other differentially regulated regions in the neural tube, including ventral domain markers NKX2.2, OLIG2 and NKX6.1, motor neuron marker ISL1 and neuronal marker TUBB3. We observed no significant changes in induction frequency of any of the fates in either actuated or unactuated conditions with MagCs compared to controls without MagCs, indicating that the presence of MagCs does not selectively abrogate fates in our model system (**Fig.5a**). Upon magnetic actuation, FOXA2+, NKX6.1+, OLIG2+ and PAX6+ hNTmOs all exhibited an increase in fate patterning frequency compared to either control organoids and unactuated hNTmOs. We further assessed the occurrence of intermediate and ventral fates within the same organoid and found a higher frequency of PAX6+/NKX6.1+ double positive organoids in actuated conditions (26.2%) compared to control (12.5%) and unactuated hNTmOs (15.8%) (**Supplementary Fig.8**), underscoring the enhanced patterning as a result of magnetic actuation.

We next assessed whether a directional patterning bias occurred in actuated hNTmOs with reference to the magnet position. To do so, for each marker we projected the fate expressions from all observed organoids onto the same space, scaling respective organoid size where necessary to ensure proper mapping while keeping track of the magnet position (**Fig.5b**). Strikingly, strong polarization of FOXA2, NKX6.1 and PAX6 were observed in actuated organoids, with preferential expression towards the magnet position for FOXA2 and NKX6.1 and away from the magnet position for PAX6. NKX6.1 expression appeared more expansive than FOXA2 floor plate expression as it known to be more broadly expressed across multiple ventral domains. NKX2.2 and OLIG2 expression was also seen across broader areas (**Fig.5b** white arrow heads), appearing laterally to the magnetic force vector. By extension, motor neuron ISL1+ cells, possibly originating from OLIG2+ progenitor motor neuron (pMN) cells, also had high bilateral expressions in actuated hNTmOs (**Fig.5b** white arrow heads). In contrast, in control and unactuated hNTmOs, integrated expressions of most fates showed random hotspots with no directional bias.

We further quantified directional patterning bias by evaluating the average fate fluorescence intensities away and towards the magnet (**Fig.5c**). We found the most significant differences to be associated with FOXA2, NKX6.1 and PAX6, which are the domains that most polarize towards or away from the magnet. Linking these results to observations of proliferation suggests that floor plate cells pattern in regions with lower proliferation while those of the intermediate fates are preferentially found in regions with higher proliferation. This domain allocation may then provide the space for other ventral domains to emerge, such as the p3 (NKX2.2) and pMN (OLIG2) domains that occupy lateral positions

To further illustrate that force localization biases not only pattern direction of a single fate, but the organization of multiple patterns and fates, we analyzed integrated FOXA2, NKX6.1, NKX2.2, OLIG2, and PAX6 expression profiles in scaled organoids along various projection angles (**Fig.5d**). We then investigated the correlation, at each projection angle, with

an ideal pattern that minimally recapitulates the dorsoventral organization of the same domains in the in-vivo neural tube. Fates in the control and unactuated hNTmO cases occurred largely at random locations, therefore, as expected, rotation of the organoids did not yield strong correlation with the ideal pattern (**Fig.5d**). In the magnetically actuated hNTmOs, a strong correlation with the ideal pattern was observed at a rotation angle of 0°, which is the baseline orientation with reference to the magnet position (**Fig.5d**). This analysis demonstrated that pattern organization is a function of magnet position and that this organization is not recapitulated in magnet-free and control conditions at any other angle. To better visualize pattern organization, we merged the top 20% of the integrated fluorescence intensity of FOXA2, NKX6.1, OLIG2 and PAX6 expressions within the same organoid space keeping the original organoid orientation (**Fig.5e**). Compared to the control and unactuated hNTmOs, actuated hNTmOs showed clear dorsoventral organization with mutual exclusion of FOXA2, OLIG2 and PAX6 domains in the correct order, with NKX6.1 sharing the domain space with FOXA2 and OLIG2 but not PAX6. Although not all magnetically actuated organoids yield this degree of organization due to variations between organoids, these results demonstrate that fate specification follows a domain allocation program that is determined by the magnetic force acting on the organoid. In addition, the spatially encoded fate specifications we observe as a function of force stimulation suggest that that local forces may act as a mechanical frame of reference to orient and guide patterns across the tissue.”

We have also updated **Figure 5** in the manuscript.

Figure 5: Local magnetic actuation biases patterning in hNTmOs. **a** Representative images showing FOXA2, NKX6.1, NKX2.2, OLIG2, ISL1, TUBB3, and PAX6 expressions in day 11 hNTmOs under various actuation conditions. Respective quantification of fate induction and patterning in hNTmOs under various actuation condition (n = 3 independent experiments for NKX6.1, NKX2.2, OLIG2, ISL1, TUBB3, n = 4 independent experiments for FOXA2, n = 5 independent experiments for PAX6 except for the No Mag condition n = 4 independent experiments, a total of >60 hNTmOs for all data points per condition). Statistical analysis was determined by unpaired two-sided t-test FOXA2:Mag-Control $p = 0.0092$, FOXA2: Mag-No Mag $p = 0.0143$,

NKX6.1:Mag-Control $p = 0.0040$, NKX6.1: Mag-No Mag $p = 0.0317$, OLIG2:Mag-Control $p = 0.0162$, PAX6:Mag-Control $p = 0.085$, PAX6:Mag-No Mag $p = 0.0467$. **b** Integrated fate expressions in scaled organoids color coded by expression frequency. Hotspots in the resultant heatmaps convey regions with the highest frequency of fate expression across all analyzed organoids compared to cooler regions with less frequent expressions. White arrows denote OLIG2 and ISL1 lateral expression hotspots (data from **a**) **c** Schematic representation of biased fate expressions under magnetic actuation condition. Quantification of FOXA2, NKX6.1, NKX2.2, OLIG2, ISL1, and PAX6 intensity ratio for day 11 control organoids and hNTmOs under various actuation conditions (data from **a**). FOXA2:Mag-Control $p = 0.0074$, FOXA2: Mag-No Mag $p = 0.0104$, NKX6.1:Mag-Control $p = 0.0051$, NKX6.1: Mag-No Mag $p = 0.0039$, NKX2.2: Mag-No Mag $p = 0.0386$, ISL1:Mag-Control $p = 0.0205$, ISL1: Mag-No Mag $p = 0.0488$, PAX6:Mag-Control $p = 0.0423$, PAX6:Mag-No Mag $p = 0.0020$. **d** Schematic representation of organoid rotation and expression profile assessment of domains along the dorsoventral (D-V) axis and their correlation compared to an ideal pattern. Correlation heatmap at various organoid rotation angles color-coded by correlation strength. **e** Highest 20% of integrated fluorescent intensity of fate expression from **b** for FOXA2, NKX6.1, OLIG2 and PAX6 and the associated expression profile. Scalebar 50 μm . Error bars are SEM.

We have updated the “**Fate specification and patterning studies**” section in the “**Materials and methods**”, below is an excerpt from the manuscript.

“To interrogate expression phenotypes for each of the markers, the marker expression image was scaled and projected onto a circular hypothetical organoid while retaining the original orientation. For each marker, the hypothetical organoid images were projected onto the same space and the intensities were integrated. Intensity variation was then color-coded. Higher intensities revealed regions with highest marker positional bias in the organoid space across 50 to 100 organoids from 3 to 5 biological replicates. To quantify the marker directional bias upon internal force generation, inverted microscopy images were assessed, in a similar manner as for EdU stains, for each of the markers separately.

To investigate pattern organization with reference to the magnetic field direction, we evaluate the projected intensity profiles of FOXA2, NKX6.1, NKX2.2, OLIG2 and PAX6 of the integrated marker expression every 30° from -90° to 90° from the original organoid orientation. The profiles were first normalized then compared to an ideal square intensity pattern and a correlation was established using the `corrcoef` function MATLAB (R2018a, The MathWorks Inc.). A correlation heatmap was generated using the R package `heatmap.2`. To visualize the enhanced pattern organization in magnetoids we merged the top 20% of the integrated expressions of FOXA2, NKX6.1, OLIG2 and PAX6 fates and reported their profiles at the original organoid orientation.”

Reviewer 3

- 1) The authors showed magnetoid generation (fig1), the actuation of magnetic force in organoid (figs 2-3), the local actuation affects growth and proliferation in organoids (fig4), and the local actuation induce tissue patterning of floor plate/intermediate zone (fig5). It looks the logical flow of fig1-4 is smooth, but there still exist some gaps between figs1-4 and fig5. Why biased growth/proliferation induce regional patterning? How magnetic forces trigger the patterning? What types of signal cascade work during the magnetic actuations. These questions are still exist, and authors can fill the gaps between fig4 and 5 by answering these questions, and make clear the correlation between each figure.

In-vivo studies have previously shown that proliferation is higher in intermediate domains of the developing neural tube compared to ventral domains, resulting in an expansion of intermediate domains (Bonnet, F. *et al.* Neurogenic decisions require a cell cycle independent function of the CDC25B phosphatase. *Elife* **7**, e32937 (2018) and Ranga, A. *et al.* Neural tube morphogenesis in synthetic 3D microenvironments. *Proc. Natl. Acad. Sci. U. S. A.* **113**, E6831–E6839 (2016)). This correlation between domain specification and cell cycle suggests that mechanical modulation of cell cycle using magnetic force could lead to changes in cell fate specification and patterning, which we set out to explicitly test in our in-vitro experiments. We first began by looking at cell cycle and observed a reduction of cell proliferation in the compressed regions of the hNTmOs as reported by the EdU staining in **Figure 4b**. We next observed that FOXA2 patterning is biased towards regions of low proliferation. Therefore, compressing cells decreases proliferation as well as the ability to assume intermediate identities (PAX6) but increases the likelihood of adopting ventral fates such as FOXA2. Indeed we have observed and reported this in a previous publication (Abdel Fattah *et al.*, *Nature Communications*, 2021), where stretching human neural tube organoids led to an increase in proliferation that correlated with a reduction in FOXA2 expressing cells. In those experiments, enhanced pattern formation was seen, but in arbitrary directions due to the global force applied. Here, and in a complementary way, we show that local reductions in proliferation caused by local mechanical compression biases FOXA2 expression in these regions to form a pattern. We therefore believe that the results of this study, in a complementary and more targeted way, indicate that cell cycle is a driver of fate specification and subsequent patterning. To make this link clearer to the reader, we have therefore incorporated the reviewer's suggestion and have revised the "**Local magnetic actuation biases growth and proliferation in human neural tube magnetoids**" section:

"Increased cell cycle and proliferation have been reported in the intermediate domains of the developing mouse neural tube, compared to ventral domains³³, where the floor plate cells remains non-proliferative¹³. Moreover, mechanical stresses have been shown to guide cell fate specification and patterning in mouse NTOs^{13,34}. We have previously demonstrated that global mechanical stretching increased cell proliferation in human NTOs, concomitant with a reduction in FOXA2+ cells, thereby leading to enhanced patterning of this subpopulation¹⁶. Taken together, these results suggest that mechanics-induced modulation of cell cycle in hNTmOs could be be linked to changes in fate specification and patterning. We therefore thought to investigate whether the observed changes in local proliferation in our hNTmOs were linked to mechanically-driven biases in patterning."

To further emphasize the important impact of local forces, we have significantly expanded our analysis and experiments to include more ventral domains and additional neuronal identities, which we detail in response to comment 2 below.

- 2) *Relating to point 1, the data in fig5 need more detailed analysis. The authors showed floor plate and intermediate domains in neural tube organoids but the expression of ventral domain markers can be added for better interpretation of the data. There are some gap between expression of FOXA2 and PAX6 in fig5c, and there will exist ventral domain. Besides, the authors showed FOXA2 and PAX6 expression separately, but it is better to show in the same samples.*

We thank the reviewer for these excellent suggestions. We agree with the extensive characterization of other fates and generated a comprehensive map of these organoid domains with respect to magnet location to emphasize the role of local mechanical forces in our model system. We have therefore conducted an extensive additional set of experiments to characterize additional ventral domains NKX2.2 (p3), OLIG2 (pMN), NKX6.1 (p3-pMN-p2) as well as motor neuron markers ISL1 and pan neuron marker TUBB3. The results of these experiments are detailed in the updated **Figure 5**, where we assessed patterning of each of these markers. To gain an overview of the overall patterning morphology and directional bias, for each fate we projected the fate expressions from all observed organoids onto the same space, scaling respective organoid size where necessary to ensure proper mapping while keeping track of the magnet position. This resulted in a heatmap of integrated expression for each fate (**Figure 5b**). The results clearly demonstrates hotspots of fate expression, which convey the most likely regions a fate will be expressed within the organoid. Of interest is the magnetically actuated condition where FOXA2 and NKX6.1 are primarily expressed in the organoid half closest to the magnet (less proliferative and compressed regions), contrasting with the expression of PAX6, which primarily occurs on the opposite half (more proliferative regions facing away from the magnet). Therefore, we show, as suggested by the reviewer, that the FOXA2 and PAX6 domains are mutually exclusive when magnetically actuated. Strikingly, we also note that OLIG2 and ISL1 expressions flank the hNTmO centerline bilaterally, in an expression pattern similar to their in-vivo counterpart. Comparatively, in the control and unactuated hNTmO conditions, although fate hotspots do occasionally occur, all fates and particularly FOXA2, NKX6.1 and PAX6 form expansive domains that are in sharp contrast to the magnetic actuation case where domains are spatially defined.

We have quantified these biases in expression towards or away from the magnet side for each marker as we previously did for FOXA2 and PAX6. This was done by measuring the marker intensity in the two halves of the organoids (facing towards and away from the magnet) and reporting the ratio of these two values (**Figure 5c**). We found the most significant difference to be associated with FOXA2, NKX6.1 and PAX6, which are the domains that most polarize towards or away from the magnet.

To further illustrate the pattern direction correlation with force localization, we analyzed the integrated FOXA2, NKX6.1, NKX2.2, OLIG2 and PAX6 expressions in scaled organoids along various projection angles (**Figure 5d**). We chose these transcription factors because they represent ventral domain patterning along the dorsoventral axis. We then compared, at each projection angle, the correlation with an ideal pattern that minimally recapitulates the ventral organization of the in-vivo neural tube. We divided this ideal pattern into 4 domains focusing on the ventral side 1) the floor plate (FOXA2), 2) p3 domain (NKX2.2) 3) progenitor motor neuron domain (OLIG2) and 4) intermediate/dorsal domains (PAX6), which are mutually exclusive, and 5) NKX6.1, which colocalizes with FOXA2, NKX2.2, and OLIG2. Fates in the control and unactuated hNTmO cases occurred largely at random locations, therefore, as expected, rotation of the organoids along 180° did not yield strong correlation with the ideal pattern. In the magnetically actuated hNTmOs, a strong correlation with the ideal pattern was observed at a rotation angle of 0°, which is the unaltered orientation with reference to the magnet position. This analysis demonstrated that pattern organization is a function of magnet position and that this organization is not recapitulated in magnet-free and control conditions at any other angle.

To further illustrate this force induced organization, we combined the top 20% of the integrated fluorescence intensity of FOXA2, NKX6.1, OLIG2 and PAX6 expressions and reported the analyzed expressions along a rotation angle of 0°. Although this is an integrated expression that is accumulated over many organoids, the results clearly demonstrate the enhanced pattern organization

in magnetic simulation conditions. These results suggest that although not all the fates are recapitulated in each organoid, on average, in magnetic actuation conditions, fates are restricted to certain regions. Here FOXA2 emerges in the ventral regions (towards the magnet) where it is largely mutually exclusive with OLIG2 expression, which flanks the lateral sides, similar to *in vivo* observations. While NKX6.1 clearly overlaps with FOXA2 and OLIG2 domains, it is mutually exclusive with PAX6 expression, which is restricted to the dorsal-most (away from the magnet) side of the reconstructed organoid. Control and unactuated hNTmOs clearly do not exhibit similar level domain organization, emphasizing the importance of force localization. Local forces can thus be considered as a mechanical frame of reference in organoids to orient and guide patterns.

We have revised the “**Local magnetic actuation guides tissue-wide patterning in human neural tube magnetoids**” section to reflect these new experimental observations and analyses:

“We first investigated the effect of the magnetic field alone on the induction of ventral and intermediate neural tube domains by analysing expression of FOXA2 and PAX6 respectively. When subjecting MagC-free organoids to a magnetic field, we found that cytoarchitecture and frequency of FOXA2 and PAX6 were similar to control conditions not subjected to a magnetic field, indicating that magnetic fields alone do not alter the biological responses in this model system (**Supplementary Fig.7**). We further expanded our analysis to assess other differentially regulated regions in the neural tube, including ventral domain markers NKX2.2, OLIG2 and NKX6.1, motor neuron marker ISL1 and neuronal marker TUBB3. We observed no significant changes in induction frequency of any of the fates in either actuated or unactuated conditions with MagCs compared to controls without MagCs, indicating that the presence of MagCs does not selectively abrogate fates in our model system (**Fig.5a**). However, upon magnetic actuation, FOXA2+, NKX6.1+, OLIG2+ and PAX6+ hNTmOs all exhibited an increase in fate patterning frequency compared to either control organoids and to unactuated hNTmOs. We further assessed the occurrence of intermediate and ventral fates within the same organoid and found a higher frequency of PAX6+/NKX6.1+ double positive organoids in actuated conditions (26.2%) compared to control (12.5%) and unactuated hNTmOs (15.8%) (**Supplementary Fig.8**), underscoring the enhanced patterning as a result of magnetic actuation.

We next assessed whether a directional patterning bias occurred in actuated hNTmOs with reference to the magnet position. To do so, for each marker we projected the fate expressions from all observed organoids onto the same space, scaling respective organoid size where necessary to ensure proper mapping while keeping track of the magnet position (**Fig.5b**). Strikingly, strong polarization of FOXA2, NKX6.1 and PAX6 were observed in actuated organoids, with preferential expression towards the magnet position for FOXA2 and NKX6.1 and away from the magnet position for PAX6. NKX6.1 expression appeared more expansive than FOXA2 floor plate expression as it known to be more broadly expressed across multiple ventral domains. NKX2.2 and OLIG2 expression was also seen across broader areas (**Fig.5b** white arrow heads), appearing laterally to the magnetic force vector. By extension, motor neuron ISL1+ cells, possibly originating from OLIG2+ progenitor motor neuron (pMN) cells, also have high bilateral expressions in actuated hNTmOs (**Fig.5b** white arrow heads). In contrast, in control and unactuated hNTmOs, integrated expressions of most fates did not show directional bias, with random hotspots.

We further quantified directional patterning bias by evaluating the average fate fluorescence intensities away and towards the magnet (**Fig.5c**). We found the most significant differences to be associated with FOXA2, NKX6.1 and PAX6, which are the domains that most polarize towards or away from the magnet. Linking these results to observations of proliferation suggests that floor plate cells pattern in regions with growth restriction and lower proliferation while those of the intermediate fates are preferentially found in regions

with higher proliferation. This domain allocation, may then provide the space for other ventral domains to emerge, such as the p3 (NKX2.2) and pMN (OLIG2) domains that occupy lateral positions

To further illustrate that force localization biases not only pattern direction of a single fate, but the organization of multiple patterns and fates, we analyzed integrated FOXA2, NKX6.1, NKX2.2, OLIG2, and PAX6 expression profiles in scaled organoids along various projection angles (**Fig.5d**). We then investigated the correlation, at each projection angle, with an ideal pattern that minimally recapitulates the dorsoventral organization of the same domains in the in-vivo neural tube. Fates in the control and unactuated hNTmO cases occurred largely at random locations, therefore, as expected, rotation of the organoids did not yield strong correlation with the ideal pattern (**Fig.5d**). In the magnetically actuated hNTmOs, a strong correlation with the ideal pattern was observed at a rotation angle of 0°, which is the baseline orientation with reference to the magnet position (**Fig.5d**). This analysis demonstrated that pattern organization is a function of magnet position and that this organization is not recapitulated in magnet-free and control conditions at any other angle. To better visualize pattern organization, we merged the top 20% of the integrated fluorescence intensity of FOXA2, NKX6.1, OLIG2 and PAX6 expressions within the same organoid space keeping the original organoid orientation (**Fig.5e**). Compared to the control and unactuated hNTmOs, actuated hNTmOs showed clear dorsoventral organization with mutual exclusion of FOXA2, OLIG2 and PAX6 domains in the correct order, with NKX6.1 sharing the domain space with FOXA2 and OLIG2 but not PAX6. Although not all magnetically actuated organoids yield this degree of organization due to variations between organoids, these results demonstrate that fate specification follows a domain allocation program that is determined by the magnetic force acting on the organoid. In addition, the spatially encoded fate specifications we observe as a function of force stimulation suggest that that local forces may act as a mechanical frame of reference to orient and guide patterns across the tissue.”

We have also updated **Figure 5** in the manuscript.

Figure 5: Local magnetic actuation biases patterning in hNTmOs. **a** Representative images showing FOXA2, NKX6.1, NKX2.2, OLIG2, ISL1, TUBB3, and PAX6 expressions in day 11 hNTmOs under various actuation conditions. Respective quantification of fate induction and patterning in hNTmOs under various actuation condition (n = 3 independent experiments for NKX6.1, NKX2.2, OLIG2, ISL1, TUBB3, n = 4 independent experiments for FOXA2, n = 5 independent experiments for PAX6 except for the No Mag condition n = 4 independent experiments, a total of >60 hNTmOs for all data points per condition). Statistical analysis was determined by

unpaired two-sided t-test FOXA2:Mag-Control $p = 0.0092$, FOXA2: Mag-No Mag $p = 0.0143$, NKX6.1:Mag-Control $p = 0.0040$, NKX6.1: Mag-No Mag $p = 0.0317$, OLIG2:Mag-Control $p = 0.0162$, PAX6:Mag-Control $p = 0.085$, PAX6:Mag-No Mag $p = 0.0467$. **b** Integrated fate expressions in scaled organoids color coded by expression frequency. Hotspots in the resultant heatmaps convey regions with the highest frequency of fate expression across all analyzed organoids compared to cooler regions with less frequent expressions. White arrows denote OLIG2 and ISL1 lateral expression hotspots (data from **a**) **c** Schematic representation of biased fate expressions under magnetic actuation condition. Quantification of FOXA2, NKX6.1, NKX2.2, OLIG2, ISL1, and PAX6 intensity ratio for day 11 control organoids and hNTmOs under various actuation conditions (data from **a**). FOXA2:Mag-Control $p = 0.0074$, FOXA2: Mag-No Mag $p = 0.0104$, NKX6.1:Mag-Control $p = 0.0051$, NKX6.1: Mag-No Mag $p = 0.0039$, NKX2.2: Mag-No Mag $p = 0.0386$, ISL1:Mag-Control $p = 0.0205$, ISL1: Mag-No Mag $p = 0.0488$, PAX6:Mag-Control $p = 0.0423$, PAX6:Mag-No Mag $p = 0.0020$. **d** Schematic representation of organoid rotation and expression profile assessment of domains along the dorsoventral (D-V) axis and their correlation compared to an ideal pattern. Correlation heatmap at various organoid rotation angles color-coded by correlation strength. **e** Highest 20% of integrated fate expression from **b** for FOXA2, NKX6.1, OLIG2 and PAX6 and the associated expression profile. Scalebar 50 μm . Error bars are SEM.

We have updated the “**Fate specification and patterning studies**” section in the “**Materials and methods**”, below is an excerpt from the manuscript.

“To interrogate expression phenotypes for each of the markers, the marker expression image was scaled and projected onto a circular hypothetical organoid while retaining the original orientation. For each marker, the hypothetical organoid images were projected onto the same space and the intensities were integrated. Intensity variation was then color-coded. Higher intensities revealed regions with highest marker positional bias in the organoid space across 50 to 100 organoids from 3 to 5 biological replicates. To quantify the marker directional bias upon internal force generation, inverted microscopy images were assessed, in a similar manner as for EdU stains, for each of the markers separately.

To investigate pattern organization with reference to the magnetic field direction, we evaluate the projected intensity profiles of FOXA2, NKX6.1, NKX2.2, OLIG2 and PAX6 of the integrated marker expression every 30° from -90° to 90° from the original organoid orientation. The profiles were first normalized then compared to an ideal square intensity pattern and a correlation was established using the `corrcoef` function MATLAB (R2018a, The MathWorks Inc.). A correlation heatmap was generated using the R package `heatmap.2`. To visualize the enhanced pattern organization in magnetoids we merged the top 20% of the integrated expressions of FOXA2, NKX6.1, OLIG2 and PAX6 fates and reported their profiles at the original organoid orientation.”

We found that FOXA2 and PAX6 expressions were not often co-expressed within the same organoid. To make it clearer for the reader we have assessed the percentage of organoids that are double positive for ventral (NKX6.1, since it encompasses more ventral fates than just the floor plate) and intermediate fates (PAX6), finding them to vary from 12 to 15% in control and unactuated hNTmOs but reach up to 26% in actuated hNTmOs. However since the percentage of double positive organoids remains relatively low, we did not want to overstate DV patterning by showing images that are not representative of our observations. To ensure that this is clear to the reader we have revised the “**Local magnetic actuation guides tissue-wide patterning in human neural tube magnetoids**” section, below is an excerpt from the manuscript.

“We further assessed the occurrence of intermediate and ventral fates within the same organoid and found a higher frequency of PAX6+/NKX6.1+ double positive organoids in actuated conditions (26.2%) compared to control (12.5%) and unactuated hNTmOs (15.8%) (**Supplementary Fig.8**), underscoring the enhanced patterning as a result of magnetic actuation.”

We have also included the figure below (**supplementary figure 8**) in the manuscript.

Supplementary Fig.8: Representative images for ventral (NKX6.1) and intermediate (PAX6) double positive day 11 organoids under various actuation conditions (n = 3 independent experiments from **Fig. 5a**). Quantification of double positive organoids for various actuation conditions (n = 3 independent experiments from **Fig.5** for a total of >60 organoids per condition). Statistical analysis was determined by unpaired two-sided t-test Mag-Control $p < 0.0308$. Scalebar 50 μm . Error bars are SEM.

We have updated the “**Fate specification and patterning studies**” section in the “**Materials and methods**”:

“To evaluate the percentage of dorsoventral patterning we assessed the expression of NKX6.1 (ventral fates) and PAX6 (intermediate fates) in the same organoid and reported the overall fraction of double positive organoids.”

3) *The expression of PAX6 decrease in fig5 b No mag, but there is not interpretation about this data. Besides, the staining of PAX6 in supplementary fig6 is not clear.*

We have added an additional replicate to the PAX6 experiments for induction and patterning assessment in order to investigate this decrease. We found that the additional induction and patterning data were in line with expected results and with the two other replicates suggesting that the one replicate with no patterning was likely an outlier. We nevertheless considered it in our statistical comparisons but pointed out to the reader that this particular data point is likely an outlier.

We have updated the “**Fate specification and patterning studies**” section in the “**Materials and methods**”, below is an excerpt from the manuscript.

“We note that one of the biological replicates in the unactuated hNTmO cases (**Fig.5a**) yielded no patterning of PAX6 expressions. We believe this to be an outlier as all other 3 biological replicates cluster together.”

Below is part of **Figure 5a** that points to the additional replicate that we highlight in red

Part of Figure 5a. PAX6 induction and patterning quantification

We have also revised PAX6 as well as FOXA2 expressions in Supplementary Figure 6 (Now **Supplementary Figure 7**) to have clearer PAX6 expressions. Below is the revised figure.

Supplementary Fig. 7: Representative images phalloidin, FOXA2 and PAX6 expressions in day 11 MagC-free hNTOs under control and magnetic conditions (n = 3 independent experiments). Induction rates of FOXA2 and PAX6 in hNTOs cultured under control and magnetic conditions (n = 3 independent experiments for a total of 90 hNTOs per condition). Scalebar 50 μ m. Error bars are SEM.

We thank the reviewer for the following minor comments.

4) Is it possible to explain more detail about how MagCs forms rod shaped structure.

To make it clearer to the reader and avoid confusion, we expanded the description of rod shaped MagC generation the manuscript, below is an excerpt from the **“Magnetoid generation”** section.

“Magnetized pluripotent aggregates with localized MagCs (**Fig.1a**) were produced by first centrifugating mhPSCs followed by a second centrifugation step to aggregate the non-magnetic hPSCs. This consecutive centrifugation allows mhPSCs and their MagC tags to collect together at the bottom of the aggregation well. We subsequently perform a 1 day incubation under a magnetic field. Because the MagCs are only transiently attached to their cellular chaperones, once they become magnetized under the magnetic field, a force is generated within the MagCs and they leave the cell surfaces in an attempt to travel towards the magnet. However, because MagCs are in close proximity to each other, they undergo directed assembly as they interact and influence each other through their own induced magnetic fields, allowing them to build rod shaped MagCs inside the organoids which ensures force localization. Instantly mixing magnetic nanoparticles with non-magnetized hPSCs produced magnetoids with multiple MagC clusters and thus force foci, which is an undesirable effect that diffuses the force rather than localizes it. These magnetized pluripotent aggregates were then embedded in PEG hydrogels with a stiffness of 2 kPa as previously described¹⁶, and could

then be differentiated under constant or periodic magnetic actuation to impose internal local forces through the displacement of the rod-shaped MagCs”

- 5) *I have concerns about the naming of “magnetoid”. The term “organoid” means organ-like tissues, then “magnetoid” means magnet like tissue and it looks the naming does not reflect the data.*

We apologize for this confusion, the term “magnetoid” is intended to be a portmanteau between the terms magnetized and organoid and it was not our intention to convey that the tissue itself is acting as a magnet or a magnetizable material. We hope that this explanation clarifies the term.

- 6) *The explanation of supplementary fig7 can be changed for better readability. The correlation of magnetic log and nucleus-MagC distance is difficult to understand for me. Please add more explanation in figure legend or text.*

We thank the reviewer for this observation. We have revised our analysis, added more data points and changed the figure such that it communicates the force, which is easier to understand than the log of the force. In this graph our intent is to show, for each quantified organoid, the distance of the cell furthest from the MagC which expresses PAX6+ (intermediate domain) or FOXA2+ (floor plate), and to therefore demonstrate that high forces, produced by larger MagCs or magnetoids that are closer to the magnet, generally result in more expansive domains of FOXA2 or PAX6 (i.e. larger distances between the domain edge (furthest cell) and the MagC). To ensure that this is clear to the reader we have revised the **“Local magnetic actuation guides tissue-wide patterning in human neural tube magnetoids”** section, below is an excerpt from the manuscript.

“Because not all MagCs have the same volume and not all the organoids are at the same distance away from the magnet surface, the generated local forces are not the same across organoids. The resultant range of forces provided us with an opportunity to examine force magnitude and the distance of cells expressing different fates. By measuring distance between the furthest expressing FOXA2+ or PAX6+ cell to the MagC in actuated magnetoids we can describe the range of the generated local force in influencing cells away from the MagC (**Supplementary Fig.10**). Overall, we found that the higher the force, the more distant the expression of the furthest FOXA2+ and PAX6+ cells (**Supplementary Fig. 10**), indicating that higher forces have larger effective areas in which cells can be influenced to adopt specific fates, and more broadly that the position of cells in relation to the source of the mechanical force is involved in specifying its fate.”

We have also revised Supplementary Figure 7 (Now **Supplementary Figure 10**) to reflect these changes:

Supplementary Fig.10: Schematic representation of biased FOXA2 and PAX6 expressions under magnetic actuation condition. Correlation of the magnetic force generated by a MagC and the distance separating the MagC and the nucleus expressing FOXA2 or PAX6 fates (n = 4 independent experiments, data from **Fig.5a**, for a total of 140 organoids per fate).

- 7) *There are some overstatements. Abstract line12~: “This versatile approach, ~” Authors showed neural tube patterning, and the descriptions can be focused on the authors data.*

We have revised the abstract to particularly mention neural tube organoids. below is an excerpt from the Abstract.

“This approach, uniquely enabled by nanoparticle technology, allows for precise and locally controllable mechanical actuation in human neural tube organoids, and could be widely applicable to interrogate the role of local mechanotransduction in developmental and disease model systems.”

- 8) *Page8 line25~: “suggesting that bending of the ~~~~”. This interpretation need in vivo corresponding data.*

We have taken out this sentence from the manuscript.

- 9) *Page10 line21~: “Moreover, while we demonstrate ~~~~”. Authors showed neural tube patterning, and the descriptions can be focused on the authors data.*

We have revised this line to particularly mention neural tube organoids. Below is an excerpt from the manuscript.

“Moreover, while we demonstrate the use of local magnetic actuation in human neural tube organoids, we believe that the use of magnetoids in other multicellular contexts could serve to address a wide variety of mechanobiological questions related to the effects of local forces in development and disease.”

REVIEWERS' COMMENTS

Reviewer #1 (Remarks to the Author):

The manuscript has been properly revised and I am convinced with all answers and modifications.

Reviewer #2 (Remarks to the Author):

The revised manuscript by Fattah and colleagues includes significant additional experiments that raises this reviewer's enthusiasm for the work. In particular, the addition of Figure 5 addresses several of my prior criticisms and I applaud the authors' attentive revisions and rebuttal to initial critiques. There are several minor grammatical issues that hopefully will be fixed by the journal's editing team, but overall, I am supportive of publication in Nature Communications.

Reviewer #3 (Remarks to the Author):

The authors responded well to reviewer's comments. Especially, the revised figure5 become very interesting one.

One minor comment:

Please add information of immunostaining protocol with the information of antibody (company, cat number, concentration).

Reviewer 1

1) The manuscript has been properly revised and I am convinced with all answers and modifications.

We thank the reviewer for supporting our revised work.

Reviewer 2

1) The revised manuscript by Fattah and colleagues includes significant additional experiments that raises this reviewer's enthusiasm for the work. In particular, the addition of Figure 5 addresses several of my prior criticisms and I applaud the authors' attentive revisions and rebuttal to initial critiques. There are several minor grammatical issues that hopefully will be fixed by the journal's editing team, but overall, I am supportive of publication in Nature Communications.

We thank the reviewer for supporting our revised work. We have revised the manuscript for grammatical errors.

Reviewer 3

1) The authors responded well to reviewer's comments. Especially, the revised figure5 become very interesting one.

We thank the reviewer for supporting our revised work.

2) Please add information of immunostaining protocol with the information of antibody (company, cat number, concentration).

We have supplemented the Methods section with antibody information. Below is an excerpt from the revised **Methods** section.

“First organoids were fixed with 4% PFA for 2 h followed by PBS washing and then permeabilization and blocking using a solution of 0.3% Triton X (PanREAC AppliChem) and 0.5% BSA (Sigma-Aldrich) for 30 min. Next, primary antibodies suspended in blocking and permeabilization solution were applied to the samples for 24 h. The following primary antibodies were used in this study, FOXA2 (Santacruz (sc-374376), mouse monoclonal, dilution 1:200), NKX2.2 (DSHB (DSHB - 74.5A5), mouse monoclonal, dilution 1:200), OLIG2 (R&D systems (AF 2418), goat polyclonal, dilution 1:200), NKX6.1 (DSHB (DSHB - F55A10), mouse monoclonal, dilution 1:200), ISL1/2 (DSHB (DSHB - 39.4D5), mouse monoclonal, dilution 1:200), TUBB3 (Biolegend (802001), rabbit polyclonal, dilution 1:200), PAX6 (Biolegend (901301), rabbit polyclonal, dilution, 1:200). The samples were subsequently

washed three times using PBS over a period of 24 h. Immunolabeling was then performed using secondary antibodies suspended in blocking and permeabilization solution for an additional 24 h. The following secondary antibodies were used in this study, Alexa Fluor 555 (Invitrogen (A-31570), donkey polyclonal, donkey anti mouse, dilution 1:500), Alexa Fluor 647 (Invitrogen (A-31573), donkey polyclonal, donkey anti rabbit, dilution 1:500), Alexa Fluor 647 (Invitrogen (A-21447), donkey polyclonal, donkey anti goat, dilution 1:500). The samples were subsequently washed three times using PBS over a period of 24 h before imaging.”